# Structural Adaptations to Saline Stress: Histomorphological Changes in the Osmoregulatory and Metabolic Organs of *Perca schrenkii* Under Acute and Chronic Challenges

**DOI:** 10.3390/biology14121775

**Published:** 2025-12-11

**Authors:** Guanping Xing, Kaipeng Zhang, Shixin Gao, Yichao Hao, Zhulan Nie, Jie Wei, Tao Ai, Shijing Zhang, Jiasong Zhang, Zhaohua Huang

**Affiliations:** 1College of Life Science and Technology, Tarim University, Alar 843300, China; xingguanping1111@163.com (G.X.); zkp19980604@163.com (K.Z.); follow_spot@163.com (S.G.); 10757242166@stumail.taru.edu.cn (Y.H.); 2Xinjiang Production & Construction Corps Key Laboratory of Protection and Utilization of Biological Resources in Tarim Basin, Alar 843300, China; 3Xinjiang Production and Construction Corps Aquaculture Technology Promotion General Station, Urumqi 830002, China; taoai202506@163.com; 4Yutian Fengze Technology Aquatic Products Co., Ltd., Yutian County 848400, China; 17797917561@163.com (S.Z.); jiasongzhang@hotmail.com (J.Z.); 5Aral Changxin Fishery Co., Ltd., Aral 843300, China; huangzhaohua1968@163.com

**Keywords:** *Perca schrenkii*, salinity stress, histomorphology, osmoregulation, gill remodeling, chloride-type saline water

## Abstract

Freshwater scarcity has driven the exploration of saline–alkaline aquaculture, and Xinjiang-native *Perca schrenkii* shows high potential for such cultivation. However, the adaptive mechanisms of its organ structures to salt stress remain unclear. This study examined microscopic structural changes in the gills, kidneys, intestines, and liver of Perca schrenkii under acute and chronic salt stress: high salinity caused significant organ damage, while long-term mild salinity induced adaptive remodeling (e.g., intestinal muscle thickening, gill structure adjustment). As this is the first study to quantify these adaptations and define the safe/tolerable salinity ranges for Perca schrenkii, it provides key guidance for its use in saline–alkaline aquaculture and sustainable aquaculture expansion.

## 1. Introduction

The escalating global freshwater scarcity poses a pivotal constraint on the sustainable development of aquaculture [1,2]. With rising water stress and the growing contribution of aquaculture to global food security, the exploitation of alternative water resources has become imperative [3,4]. In arid regions such as Northwest China, chloride-type saline–alkaline water, characterized by high salinity but moderate pH, represents a vast and underutilized resource [5,6,7]. Developing aquaculture in such environments not only alleviates competition for freshwater but also offers a pathway for ecological and economic co-benefits.

The endemic Ili perch (*Perca schrenkii*) is currently in the stage of “germplasm conservation + small-scale experimental cultivation”, and is not in use in commercial aquaculture; thus, publicly available commercial yield data are lacking. Nevertheless, this species remains a highly promising candidate for aquaculture in local saline–alkaline/cold-water environments. Its value is supported by its temperature tolerance range of 1–30 °C, salinity adaptability of 5–9 ppt, and artificial aquaculture adaptability, as verified in small-scale experiments in Yining and other areas relying on the Xinjiang cold water fish industry technology system. As an endemic fish species in the Ili River basin, *Perca schrenkii* has shown considerable potential for cultivation in chloride-type saline–alkaline waters [8]. Preliminary ecological surveys and our own preliminary data suggest that *Perca schrenkii* possesses a high salinity tolerance, with a 96 h median lethal concentration (LC_50_) of 12.396 parts per thousand (ppt) and a demonstrated capacity for long-term acclimation to salinities up to 7 ppt. This tolerance is higher than that of freshwater populations of its congeneric Perca fluviatilis, which can only tolerate a maximum salinity of 10 ppt acutely. However, physiological adaptations are invariably underpinned by structural modifications [9,10]. Significant knowledge gaps persist regarding the histopathological and morphometric responses of *Perca schrenkii* to saline stress, which are crucial for a holistic understanding of its adaptation mechanisms. These mechanisms ultimately manifest in the structure of key osmoregulatory [11] and metabolic organs [12,13].

In fish, the gill [14], kidney [15], intestine [16], and liver [17] are central to maintaining homeostasis under osmotic challenge. The gill serves as the primary site for ion exchange, where lamellar structure and chloride cell proliferation directly dictate osmoregulatory efficiency [18]. The kidney is vital for ion and water reabsorption, with its structural integrity reflecting excretory capacity [19]. The intestine plays a key role in water and ion balance, its functional state mirrored in the mucosal fold architecture and goblet cell abundance [20]. Meanwhile, the liver, as the metabolic hub, often exhibits structural changes in response to systemic stress [21]. Critically, while physiological assays capture transient functional states, histomorphological examination reveals the enduring structural imprint of damage, repair, and adaptation, providing a direct and tangible record of organismal health and long-term viability [22].

Therefore, this study was designed to systematically characterize the histomorphological responses of *Perca schrenkii* to salinity stress. Guided by a preliminary determination of the 96 h median lethal salinity (LC_50_), acute stress experiments were conducted at salinities of 12, 13, and 14 ppt to encompass the critical lethal threshold. Furthermore, a safe concentration derived from the acute toxicity data informed the selection of salinity levels (3, 5, and 7 ppt) for a 60 day chronic acclimation trial. Through these controlled experiments, we quantitatively assessed morphometric alterations in four key organs—gill, kidney, intestine, and liver. By systematically analyzing *Perca schrenkii*’s histomorphological responses, this study aims to establish a model for how teleosts structurally adapt to salinity stress. The results are intended to define a suite of morphological indicators that can be applied to assess salinity tolerance and acclimation capacity across different species. Consequently, this work is expected to provide a foundational framework that advances the understanding of fish adaptation to saline environments and guides the selection and cultivation of other non-traditional species in saline–alkaline aquaculture

## 2. Materials and Methods

### 2.1. Experimental Fish and Acclimation

Healthy juvenile *Perca schrenkii* (body length: 12.15 ± 2.32 cm, body weight: 7.53 ± 1.68 g) were obtained from the Emin River basin (Tacheng Prefecture, Xinjiang, China). Before the experiment, fish were acclimatized for 14 days in 1000 L recirculating aquaculture systems under the following conditions: water temperature 22 ± 1 °C, pH 7.60 ± 0.35, dissolved oxygen ≥ 6.0 mg/L, ammonia nitrogen ≤ 0.3 mg/L, nitrite ≤ 0.03 mg/L, and salinity 0.65 ± 0.30 ppt. Fish were fed a commercial diet (crude protein ≥ 40%, crude lipid ≥ 8%) twice daily at 3% of their body weight. Feeding was suspended 24 h before the initiation of experiments or sampling. All experimental procedures were approved by the Animal Research Ethics Committee of Tarim University (Approval No. PB20250618003).

### 2.2. Experimental Design and Salinity Challenge

Two independent experiments were conducted using analytical grade NaCl (≥99.5%) to simulate the osmotic stress of chloride-type saline–alkaline water.

Water source: The experimental water is natural water from the upstream reservoir in Alar City, Xinjiang Uygur Autonomous Region.Salinity adjustment: Accurately weigh analytical grade NaCl (purity ≥ 99.5%) with an electronic analytical balance, add it to natural water, and stir thoroughly until completely dissolved.Calibration: Use a portable salinity meter (accuracy ± 0.1 ‰) to measure the salinity of the water body, and adjust the amount of NaCl added to stabilize the salinity of the water body to the target value. Monitor salinity daily, and if there are slight fluctuations in salinity, supplement a small amount of NaCl to ensure consistent salinity conditions in each treatment group.

Acute Salinity Stress Experiment: Based on the previously determined 96 h LC_50_ (12.396 ppt) determined in a preliminary pilot study, the pilot results showed that the survival rate was >80% in the 8–11 ppt group, decreased to 28–65% in the 12–14 ppt group, and was <10% (near complete lethality) in the 15 ppt group. Therefore, salinity levels of 0 ppt (control), 12 ppt, 13 ppt, and 14 ppt were set to capture the critical transition of *Perca schrenkii* from compensation to failure. Each group consisted of three replicates, with 15 fish per replicate (50 L tanks). Fish were exposed for 96 h, and sampling was conducted at 48 h, 72 h, and 96 h. The 14 ppt group suffered 100% mortality by 72 h; therefore, only the 48 h time point was available for this group.

Chronic Salinity Stress Experiment: To assess long-term acclimation, fish were exposed to 0 (control), 3, 5, and 7 ppt for 60 days. This salinity gradient was determined by first deriving a safe range with reference to the OECD Guidelines for Chronic Toxicity Testing, then integrating “long-term acclimation safety”. Meanwhile, this salinity range is consistent with that of brackish water aquaculture ponds in northwest China. Each group had three replicates, with 30 fish per replicate (200 L tanks). A 30% daily water exchange maintained water quality. Tissue sampling was performed at the end of the 60 day period.

Throughout both experiments, water temperature, pH, and dissolved oxygen were monitored daily. Photoperiod was maintained at 12 h light: 12 h dark.

### 2.3. Tissue Sampling and Histological Processing

At each sampling point, three fish per replicate (*n* = 9 per group) were anesthetized with 100 mg/L MS-222. Tissues of the gill, kidney, intestine, and liver were carefully dissected. Gill arches, kidney tissue, intestinal segments (middle part), and liver lobes were immediately fixed in Bouin’s solution for 24 h. After fixation, tissues were dehydrated through a graded ethanol series, cleared in xylene, and embedded in paraffin wax. Serial sections of 5 μm thickness were cut using a rotary microtome (Leica RM2235, Wetzlar, Germany) and mounted on glass slides. Sections were stained with routine Hematoxylin and Eosin (H&E) for general morphology.

### 2.4. Histomorphometric Analysis

Slides were examined under a light microscope (Nikon Eclipse E100, Tokyo, Japan) and digital images were captured with a calibrated camera system. Morphometric analysis was performed using ImageJ software (Version 1.54f, National Institutes of Health, Bethesda, MD, USA): ➀ Calibration images (400×: gill/liver; 200×: kidney/intestine, scale lines parallel to field) imported as TIFF (300 dpi, 8-bit grayscale) without preprocessing. ➁ ImageJ “Analyze > Set Scale” (100 μm input, “Global” unchecked) yielded coefficients: 400× (1 pixel = 0.125 μm), 200× (1 pixel = 0.25 μm). ➂ Batch verification (3 slides/batch); formal measurement if coefficient error ≤2%.

For each tissue sample per fish: ➀ P5 μm continuous slices prepared (first 5–8 edge slices discarded to avoid artifacts), 3 representative slices selected for analysis; ➁ Observation area standardized by magnification: 400× (0.0625 mm^2^/field) for gill/liver, 200× (0.25 mm^2^/field) for kidney/intestine; 3 non-overlapping parenchymal fields measured per slice, total area 0.5625 mm^2^ (gill/liver) and 2.25 mm^2^ (kidney/intestine) per organ per fish.

The following parameters were measured:

Gill: Lamellar length (μm), lamellar width (μm), and the number of chloride cells in a standard 100 μm length of the primary filament.

Kidney: Glomerular diameter (μm), tubular diameter (μm), tubular epithelial thickness (μm), and the number of glomeruli per field of view (200× magnification).

Intestine: Mucosal fold height (μm), muscularis thickness (μm), total mucosal surface area (calculated from fold height and density), and the number of goblet cells per 100 μm of intestinal epithelium.

Liver: Hepatocyte area (μm^2^), measured by outlining the cell boundaries of 20 randomly selected hepatocytes per fish.

### 2.5. Statistical Analysis

All data are presented as mean ± standard error (SE). Statistical analyses were performed using IBM SPSS Statistics 26.0. Data were checked for normality (Shapiro–Wilk test) and homogeneity of variances (Levene’s test). For the acute experiment, two-way analysis of variance (ANOVA) was used to analyze the effects of salinity, time, and their interaction, followed by Tukey’s post hoc test for multiple comparisons. For the chronic experiment, one-way ANOVA followed by Tukey’s test was used to compare differences among salinity groups at 60 days. A significance level of *p* < 0.05 was applied for all tests.

## 3. Results

### 3.1. Structural Responses to Acute Salinity Stress (96 H)

#### 3.1.1. Gill: Shortening and Widening of Lamellae with Non-Monotonic Change in Chloride Cells

As the core organ for osmoregulation in *Perca schrenkii*, the gill tissue exhibits the most direct response to acute salinity stress—it not only shows significant histopathological damage and morphometric changes, but these changes are also coordinately regulated by salinity level and exposure time (Two-way ANOVA, time, salinity, and interaction: *p* < 0.0001, Figure 1a–c).

Gill lamellae in the control group maintained a stable architecture with a length of approximately 120–130 μm (Figure 1A,a). Exposure to 12 ppt salinity did not induce significant changes in lamellar length across the 96 h period, with values remaining comparable to the control (*p* > 0.05, Figure 1C–E,a). In contrast, at 13 ppt, lamellar length significantly decreased to approximately 80 μm by 48 h and remained at this reduced level through 96 h, which was significantly shorter than in the 12 ppt group at the corresponding time points (*p* < 0.05, Figure 1F–H,a). The most severe reduction was observed at 14 ppt (48 h), where lamellar length plummeted to approximately 60 μm, significantly shorter than all other groups (*p* < 0.05, Figure 1B,a).

Concomitantly, lamellar width—an indicator of edema and lamellar fusion—exhibited a dose-dependent increase. While the control and 12 ppt groups maintained a stable width, a significant increase occurred at 13 ppt from 48 h onwards (*p* < 0.05, Figure 1b). This value was further elevated at 14 ppt (48 h), (*p* < 0.05, Figure 1b).

The population density of chloride cells demonstrated a non-monotonic response to acute salinity gradients. A moderate but significant increase was observed over time in the 12 ppt group. At 13 ppt, chloride cell count surged to a peak at 72 h, significantly higher than in the 12 ppt group (*p* < 0.05, Figure 1c); however, at the lethal salinity of 14 ppt, the chloride cell counts sharply declined at 48 h (*p* < 0.05, Figure 1c).

#### 3.1.2. Kidney: Glomerular and Tubular Impairment Under Osmotic Load

The kidney, which collaborates with the gills to maintain ion balance, exhibits histopathological and morphometric changes centered on renal tubules and glomeruli under acute salinity stress, and these responses are jointly regulated by salinity level and exposure time (two-way ANOVA, time, salinity, and interaction: *p* < 0.0001; Figure 2a–c).

The number of glomeruli per visual field in the control group remained stable at approximately 4–5 (Figure 2A,a). No significant reduction was observed in the 12 ppt group across the 96 h exposure compared to the control (Figure 2C–E,a). In contrast, exposure to 13 ppt led to a significant and progressive decrease in glomerular number, which fell to approximately 3 by 48 h and remained at this lower level, being significantly fewer than in the 12 ppt group at all corresponding time points (Figure 2F–H,a). The most severe loss occurred at 14 ppt (48 h), where the glomerular number dropped sharply to approximately 2, significantly lower than in the 13 ppt group (Figure 2B,a).

Concomitant changes were observed in tubular morphology. Tubular diameter, stable in the control and 12 ppt groups, exhibited a significant and substantial increase at 13 ppt, peaking at 72 h, which was significantly wider than in the 12 ppt group (Figure 2b). This tubular dilation was even more pronounced at 14 ppt (48 h) (Figure 2b). In parallel, tubular epithelial thickness demonstrated an inverse trend. While a gradual thinning was noted in the 12 ppt group, a significant and sustained decrease occurred at 13 ppt, which was significantly thinner than in the 12 ppt group (Figure 2c). This thinning was most severe at 14 ppt (48 h), with epithelial thickness dropping to 8 μm (Figure 2c).

#### 3.1.3. Intestine: Mucosal Atrophy and Muscularis Thinning Under Acute Stress

The intestine, which undertakes dual functions of mucosal barrier and nutrient absorption, exhibits salinity-dependent structural changes under acute salinity stress, and time, salinity, and their interaction have significant effects on all morphometric parameters (Two-way ANOVA, *p* < 0.0001; Figure 3a–c).

Mucosal architecture exhibited a biphasic response. At 12 ppt, mucosal fold height and total mucosal surface area significantly increased, peaking at 72 h, (Figure 3a,b). In contrast, at 13 ppt, these parameters showed an initial increase at 48 h followed by a significant decline by 96 h, resulting in values significantly lower than those in the 12 ppt group at corresponding time points (Figure 3a,b). The most severe impairment occurred at 14 ppt (48 h), where mucosal fold height and surface area plummeted (Figure 3B,a,b).

The muscularis layer exhibited a distinct pattern of change. While a slight, transient thickening was observed at 12 ppt, exposure to 13 ppt led to a significant and progressive thinning of the muscularis, which was significantly thinner than in the 12 ppt group from 48 h onwards (Figure 3c). This thinning was most extreme at 14 ppt (48 h), (Figure 3c).

Histological observation of goblet cells (Figure 3A–H) suggested a dynamic response. An apparent increase in goblet cell numbers was noted at 12 and 13 ppt, particularly at early time points. However, a marked reduction was observed at 14 ppt (48 h).

#### 3.1.4. Liver: Hepatocellular Swelling and Progressive Damage

As the metabolic center of the organism, the liver not only shows obvious histopathological damage under acute salinity stress, but hepatocyte area also changes significantly; time, salinity, and their interaction have significant effects on these metabolism-related indicators (Two-way ANOVA, *p* < 0.0001; Figure 4a).

Histopathological examination revealed a clear dose- and time-dependent progression of cellular injury. Hepatocytes in the control group displayed normal polygonal architecture with centrally located nuclei and no signs of pathology (Figure 4A). At 12 ppt, only occasional mild vacuolation was observed, with structural integrity largely maintained throughout the exposure (Figure 4C–E). In contrast, at 13 ppt, damage progressed from sparse vacuolation at 48 h (Figure 4F) to extensive vacuolation, mild blood cell infiltration, and widespread nuclear displacement by 96 h (Figure 4H). The most severe degeneration occurred at 14 ppt (48 h), characterized by severe vacuolation, extensive blood cell infiltration, and prominent nuclear displacement (Figure 4B).

Morphometric analysis of hepatocyte area (Figure 4A–H) corroborated the histological observations. Hepatocyte area exhibited a non-monotonic variation across salinity gradients: While the 12 ppt group showed a transient, non-significant increase in hepatocyte area at 72 h compared to the control (Figure 4a), exposure to 13 ppt resulted in significantly larger hepatocyte areas at all time points compared to the 12 ppt group, indicating sustained cellular swelling. The 14 ppt group exhibited the most pronounced hepatocyte swelling at 48 h, consistent with the severe pathological features observed.

### 3.2. Morphological Alterations Following Chronic Salinity Acclimation (60 Days)

#### 3.2.1. Gill: Structural Remodeling and Functional Trade-Offs After Chronic Acclimation

After 60 days of chronic salinity exposure, the gill tissue of *Perca schrenkii* actively underwent significant structural remodeling to maintain long-term osmotic homeostasis, with significant morphological differences between different salinity treatment groups (one-way ANOVA, *p* < 0.05; Figure 5).

A clear, salinity-dependent adjustment in lamellar dimensions was recorded. Lamellar length was greatest in the control group and showed a significant, progressive reduction with increasing salinity, reaching its shortest value at 7 ppt, which was significantly different from all other groups (Figure 5a,D). Conversely, lamellar width exhibited a non-linear response. It increased from the control value to a peak at 5 ppt, before decreasing again at 7 ppt (Figure 5b).

The population density of chloride cells also demonstrated a dynamic and non-monotonic response (Figure 5c). The highest count was observed at 3 ppt. However, those at higher salinities of 5 and 7 ppt were still elevated compared to the control, but significantly lower than that at 3 ppt.

#### 3.2.2. Kidney: Salinity-Dependent Glomerular Reduction and Tubular Dilation

Following 60 days of chronic salinity acclimation, the kidneys of *Perca schrenkii* attain ion regulatory compensation via targeted structural modifications to the glomeruli and renal tubules. Statistically significant differences in the related parameters were observed among the different salinity groups (one-way ANOVA, *p* < 0.05; Figure 6).

A pronounced, salinity-dependent reduction in glomerular number was recorded. The count decreased progressively from a maximum in the control group to approximately 15, 8, and 2 at 3, 5, and 7 ppt, respectively (Figure 6a). This indicates a severe and dose-dependent loss of filtering units under long-term salinity stress.

Tubular diameter demonstrated a non-linear response. It increased from the control value to a peak at 5 ppt, which was significantly wider than all other groups, before returning to control levels at 7 ppt (Figure 6b). This suggests a potential compensatory dilation in the 5 ppt group that is not sustained at the highest salinity. In contrast, tubular epithelial thickness remained stable across all treatment groups, showing no significant differences and indicating a preserved structural integrity of the tubular epithelium despite the other observed morphological changes (Figure 6c).

#### 3.2.3. Intestine: Structural Homeostasis and Muscular Thickening at Moderate Salinity

After 60 days of chronic salinity acclimation, the intestine of *Perca schrenkii* displayed distinct structural alterations, and statistically significant variations in mucosal and muscular layer-related parameters were detected across different salinity groups (one-way ANOVA, *p* < 0.05; Figure 7).

At 3 and 5 ppt, mucosal fold height and total mucosal surface area were similar to the control (Figure 7a,b). At 7 ppt, however, both parameters decreased significantly—fold height to 300 μm and surface area to 600,000 μm^2^—indicating mucosal atrophy at the highest salinity (Figure 7D,a,b).

A distinct adaptation was observed in the muscularis layer. While the control, 3 ppt, and 7 ppt groups maintained a similar muscularis thickness, the 5 ppt group exhibited a significant and specific thickening of this layer (Figure 7C,c). Furthermore, the distribution and abundance of goblet cells, as indicated by green flags in the micrographs, showed no obvious differences across all salinity groups.

#### 3.2.4. Liver: Biphasic Hepatocellular Response and Degeneration at High Salinity

After 60 days of chronic salinity acclimation, the liver of *Perca schrenkii* displayed distinct morphological alterations that were dependent on salinity, with significant differences in histopathological characteristics and hepatocyte area between groups (one-way ANOVA, *p* < 0.05; Figure 8).

Histopathological assessment revealed a progressive deterioration of hepatic structure with increasing salinity. Hepatocytes in the 3 ppt group showed only mild vacuolation, while the 5 ppt group exhibited more pronounced vacuolation accompanied by scattered erythrocyte accumulation. The most severe pathology was observed at 7 ppt, characterized by extensive vacuolation, prominent erythrocyte accumulation, and widespread nuclear displacement (Figure 8A–D).

Morphometric analysis of hepatocyte area (Figure 8A–D) revealed a notable biphasic response. Compared to the control group, hepatocyte area significantly decreased at 3 ppt. However, at 5 ppt, the area rebounded to a level comparable to the control, before increasing markedly at 7 ppt, which was significantly larger than all other groups (Figure 8a).

### 3.3. Synthesis of Histomorphological Responses Across Organs and Treatments

The integrated heatmap (Figure 9) provides a comparative overview of the significance of histomorphological alterations across all organs and salinity treatments, with color intensity corresponding to the statistical significance of the difference from the control group.

Distinct patterns of organ responsiveness were observed. Parameters associated with the gill (lamellar length/width, chloride cell count) and the intestine (mucosal fold height, surface area, muscularis thickness) consistently displayed darker coloration across a broader range of salinity treatments in both experiments, indicating a high frequency of significant changes. In contrast, several parameters of the kidney and liver showed significant differences over a narrower range of conditions, primarily at the highest salinities.

Under acute salinity stress, the extent and intensity of significant changes increased synergistically with both salinity concentration and exposure duration. At 14 ppt (48 h), nearly all measured parameters across the gill, kidney, intestine, and liver exhibited the most significant differences from the control. At 13 ppt, the number and significance of alterations increased from 48 h to 96 h. The 12 ppt treatment induced fewer and generally less significant changes.

In the chronic acclimation experiment, the overall pattern of significance was markedly different. At 3 ppt, very few parameters showed significant differences from the control. At 5 and 7 ppt, a subset of parameters, including gill lamellar length, intestinal muscularis thickness, and hepatocyte area, displayed significant differences. However, the collective magnitude and intensity of these significant changes across all organs were less pronounced than those observed under acute high-salinity stress (13–14 ppt).

## 4. Discussion

### 4.1. Structural Framework for Salinity Adaptation in Perca schrenkii

This study establishes a comprehensive histomorphological atlas of *Perca schrenkii*’s structural adaptations to chloride-type salinity stress, with core findings revealing a distinct dichotomy in responses to acute vs. chronic salinity exposure. Under acute stress, histopathological damage in the gill, kidney, intestine, and liver showed clear dose- and time-dependence, with a critical threshold for irreversible structural damage identified between 13 and 14 ppt. During chronic exposure (60 days, ≤7 ppt), *Perca schrenkii* achieved stable tissue homeostasis through profound structural remodeling. The gill and intestine exhibited the most significant morphological adjustments, while the kidney and liver displayed targeted adaptations. These results provide a structural basis for *Perca schrenkii*’s salinity tolerance and define morphological benchmarks for health assessment in saline aquaculture.

### 4.2. Gill and Kidney: Coordinated Frontline Defense and Homeostatic Regulation

Observations reveal an organ-specific division of labor in osmoregulation: the gill acts as a dynamic frontline interface, and the kidney serves as a key homeostatic regulator. The gill showed the most immediate and prominent morphological plasticity—a typical trait of teleosts under osmotic challenge [23,24]. Its salinity- and time-dependent lamellar fusion (reduced functional surface area) is a well-documented strategy to minimize passive ion influx and water loss in hyperosmotic environments [25,26], although this comes at the cost of respiratory efficiency. The biphasic response of chloride cells (hyperplasia at sublethal salinities 12–13 ppt, necrosis at lethal salinity 14 ppt) further confirms osmoregulatory effort and eventual failure [27,28,29]. These structural changes are entirely consistent with the reported surge and subsequent decline in gill sodium-potassium adenosine triphosphatase (NKA) activity under similar conditions, linking morphological adaptation directly to ionic regulation [30].

In contrast, the kidney maintained internal balance through more moderate but critical adjustments. Salinity-dependent reduction in glomerular number lowers glomerular filtration rate to conserve body water [31,32]. During chronic acclimation, significant tubular dilation at 5 ppt (with stable epithelial thickness) suggests targeted compensatory enhancement of ion reabsorption [33,34], shifting from “damage control” to “active adaptation”. Renal plasticity, together with dynamic structural changes in the gill, reflects a sophisticated systemic strategy. The gill regulates the primary ionic gradient at the environment-organism interface, while the kidney fine-tunes the internal milieu via regulated excretion and reabsorption [15,35,36,37].

### 4.3. Intestine and Liver: Strategic Adjustments in Metabolic and Barrier Functions

Structural adjustments in the intestine and liver enable *Perca schrenkii* to cope with salinity-induced metabolic and osmotic challenges. The intestine’s response reflects functional prioritization: increased mucosal surface area at acute low salinity (12 ppt) enhances water absorption to counter osmotic dehydration [38,39]. However, the most remarkable intestinal adaptation emerged during chronic acclimation at 5 ppt, characterized by a significant thickening of the muscularis [40]. This specific structural modification likely enhances intestinal motility, potentially serving to optimize the transit time of gut contents for more efficient water and ion absorption—a finely tuned, long-term adaptation that has been less commonly reported in other teleost species [41,42]. Stable goblet cell numbers across chronic treatments indicate intact mucosal barriers, preventing pathogen invasion and systemic inflammation [43,44].

Liver morphology reflects its central role in energy metabolism and detoxification. The hepatocyte swelling observed under acute stress is a classic indicator of metabolic disturbance and potential energy depletion, as the liver likely mobilizes energy reserves to support the costly process of osmotic regulation [45,46]. Chronic salinity exposure triggers a biphasic hepatocyte area response in *Perca schrenkii*: significant shrinkage at 3 ppt, recovery to control levels at 5 ppt, and pathological swelling at 7 ppt, indicating dynamic metabolic adjustment. The initial shrinkage at 3 ppt may reflect efficient metabolic downregulation or energy reserve loss, while subsequent swelling at 7 ppt signals homeostatic failure, coinciding with severe vacuolation and inflammation as reported previously [47]. This highlights the liver as a “metabolic sensor”, whose structural integrity is vital for sustaining long-term acclimation. In summary, the intestine modulates luminal osmotic balance via structural adjustments, while the liver regulates metabolic capacity to fuel systemic osmoregulation.

### 4.4. Hierarchical Framework of Structural Adaptation: From Damage to Acclimation

Integrating the histomorphological data across all organs allows for the construction of a hierarchical framework that describes the structural continuum of *Perca schrenkii’s* response to salinity stress, transitioning from damage to acclimation (Figure 9). This framework is defined by distinct salinity zones, each characterized by a specific pattern of organ involvement and structural change.

The first tier, the Safe Zone (≤3 ppt), is defined by minimal structural deviation. Dominated by the chronic 3 ppt group, most organ parameters show no significant difference from freshwater controls, indicating *Perca schrenkii* can maintain long-term structural and functional homeostasis here with negligible histological cost—establishing this salinity as a baseline for sustainable aquaculture.

The second tier, the Acclimation Zone (5 ppt), represents a critical phase of active, non-lethal remodeling. Responses here are not uniform damage but organ-specific strategic adjustments: the most telling adaptations—significant thickening of the intestinal muscularis and distinct dilation of the renal tubules—suggest a coordinated shift towards enhancing the efficiency of water and ion regulatory systems. The gill, while showing significant changes, maintains its structural integrity. This zone likely represents a state of compensated acclimation, where energetic costs are balanced against the benefits of inhabiting a wider ecological niche.

The third tier, the Tolerance Zone (7 ppt), represents the upper limit of long-term adaptive capacity. Structural compromises become more widespread, particularly severe gill lamellar length reduction and hepatic pathological swelling and degeneration. While these changes signal significant stress, 60 day survival confirms the achievement of a new, costlier structural homeostasis. This zone is defined by a trade-off-essential functions are preserved at the expense of optimal organ architecture, setting the physiological ceiling for chronic culture.

The final tier, the Lethal Zone (≥13 ppt), is marked by a systemic collapse of structural integrity under acute stress. At 13–14 ppt, the heatmap shows severe, significant alterations across all organs: gill lamellar fusion and chloride cell necrosis, glomerular loss, and hepatic degeneration collectively illustrate systemic breakdown, leading to irreversible damage and mortality.

This hierarchical framework underscores a clear gradient of organ sensitivity. The gill and intestine consistently served as the most responsive organs, their structural alterations providing early warning signals of salinity stress. In contrast, the kidney and liver have higher thresholds for significant change, functioning as more robust yet not invulnerable core regulatory organs. Synthesizing these organ-specific responses into a unified model explains *Perca schrenkii*’s exceptional euryhalinity and provides a tool to assess fish physiological state via histological examination, bridging microscopic anatomy, organismal ecology, and aquaculture practice. To more intuitively link the above salinity zoning, organ responses, and practical application value, we further condense and summarize the core characteristics of each level and the corresponding aquaculture recommendations (Table 1).

### 4.5. Implications, Limitations, and Future Perspectives

This study provides the first comprehensive histomorphological framework for understanding *Perca schrenkii*’s salinity adaptation, with implications for both basic biology and applied aquaculture. From a theoretical perspective, our findings move beyond merely documenting tolerance thresholds by elucidating the specific architectural modifications—such as intestinal muscularis thickening and renal tubular dilation—that underpin osmotic resilience, establishing a tangible link between physiological function and anatomical change to advance understanding of euryhaline fish homeostasis.

The practical implications of this work are immediate and significant. The delineated salinity zones—most notably, 5 ppt as an Acclimation Zone with beneficial structural adjustments and 7 ppt as the upper Tolerance Zone—offer evidence-based guidelines for *Perca schrenkii* culture in saline–alkaline waters. We recommend maintaining salinities at or below 5 ppt for optimal long-term health and growth, with 7 ppt recognized as a sustainable maximum. The specific morphological parameters quantified here, especially gill lamellar dimensions and intestinal fold architecture, can serve as valuable histological biomarkers for monitoring fish health and assessing acclimation status in commercial aquaculture settings.

Despite these contributions, several limitations should be acknowledged. The use of NaCl to simulate chloride-type saline–alkaline water, while effective for isolating the osmotic stress component, does not fully replicate the complex ionic composition (e.g., Ca^2+^, Mg^2+^, SO_4_^2−^) of natural waters, which may modulate salinity tolerance through ion-regulatory interactions. Furthermore, our study describes the structural outcomes of adaptation but does not elucidate the underlying molecular drivers.

These limitations point to clear directions for future research. Validation studies in authentic, multi-ion saline–alkaline water bodies are essential to confirm the observed adaptive responses under real-world conditions. Integrated transcriptomic and proteomic analyses are also needed to unravel the signaling pathways and gene expression networks governing structural remodeling, particularly the striking intestinal muscularis thickening at 5 ppt.

## 5. Conclusions

This study definitively characterizes the histomorphological basis of salinity adaptation in *Perca schrenkii*, establishing a clear continuum from acute damage to chronic acclimation. Our findings demonstrate that the species exhibits a high innate salinity tolerance, with a critical structural threshold for irreversible damage occurring between 13 and 14 ppt under acute stress. More significantly, we provide conclusive histological evidence that *Perca schrenkii* can fully acclimate to long-term exposure at salinities up to 7 ppt. This acclimation is not merely an absence of damage, but an active process of structural optimization, as exemplified by the targeted remodeling of the intestinal muscularis and renal tubules at moderate salinities.

The synthesis of multi-organ responses reveals a hierarchy of sensitivity, with the gill and intestine acting as primary sensors and the kidney and liver as robust core regulators. By translating physiological tolerance into a defined set of morphological indicators, this work provides both a fundamental understanding of fish adaptation to osmotic stress and a practical, evidence-based framework for the sustainable cultivation of *Perca schrenkii* in saline–alkaline waters, thereby contributing to the expansion of aquaculture to underutilized resources.

## Figures and Tables

**Figure 1 biology-14-01775-f001:**
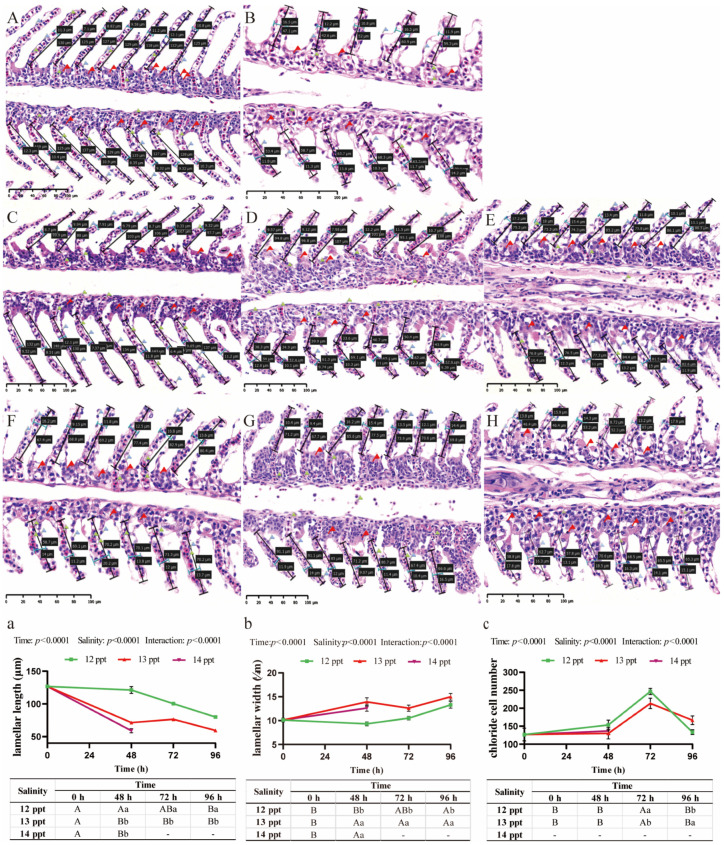
**Histopathological alterations and morphometric changes in the gill of *Perca schrenkii* under acute salinity stress**. (**A**–**H**) Representative H&E-stained gill sections. (**A**) Control group. (**B**) 14 ppt (48 h). (**C**–**E**) 12 ppt: (**C**) 48 h; (**D**) 72 h; (**E**) 96 h. (**F**–**H**) 13 ppt: (**F**) 48 h; (**G**) 72 h; (**H**) 96 h. Markers: Black double-headed arrows = lamellar length (quantified in (**a**); blue double-headed arrows = lamellar width (quantified in (**b**); red flags = chloride cells (quantified in (**c**). Scale bar = 50 μm (all micrographs). (**a**–**c**) Gill parameter dynamics (*n* = 9, mean ± SE): (**a**) lamellar length, (**b**) lamellar width, (**c**) chloride cell count. Uppercase letters = inter-temporal differences (same salinity); lowercase letters = inter-salinity differences (same time) (Two-way ANOVA, Tukey’s test, *p* < 0.05). ANOVA results (time, salinity, interaction) are shown in each panel.

**Figure 2 biology-14-01775-f002:**
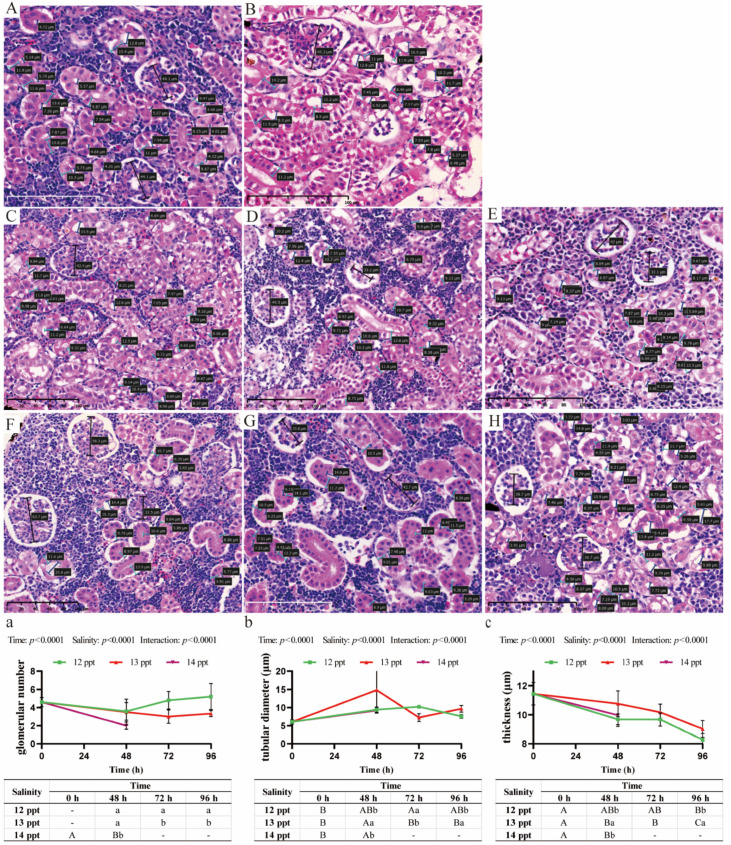
**Renal histopathological alterations and morphometric analysis in *Perca schrenkii* under acute salinity stress**. (**A**–**H**) Representative H&E-stained kidney sections. (**A**) Control group. (**B**) 14 ppt (48 h). (**C**–**E**) 12 ppt: (**C**) 48 h; (**D**) 72 h; (**E**) 96 h. (**F**–**H**) 13 ppt: (**F**) 48 h; (**G**) 72 h; (**H**) 96 h. Markers: Black double-headed arrows = glomerular longest diameter; blue double-headed arrows = tubular diameter; turquoise double-headed arrows = tubular epithelial thickness. Scale bar = 100 μm (all micrographs). (**a**–**c**) Renal morphometric parameter analysis (*n* = 9, mean ± SE): (**a**) glomerular number per field, (**b**) tubular diameter, (**c**) tubular epithelial thickness. Uppercase letters = inter-temporal differences (same salinity); lowercase letters = inter-salinity differences (same time) (Two-way ANOVA, Tukey’s test, *p* < 0.05). ANOVA results (time, salinity, interaction) are shown in each panel.

**Figure 3 biology-14-01775-f003:**
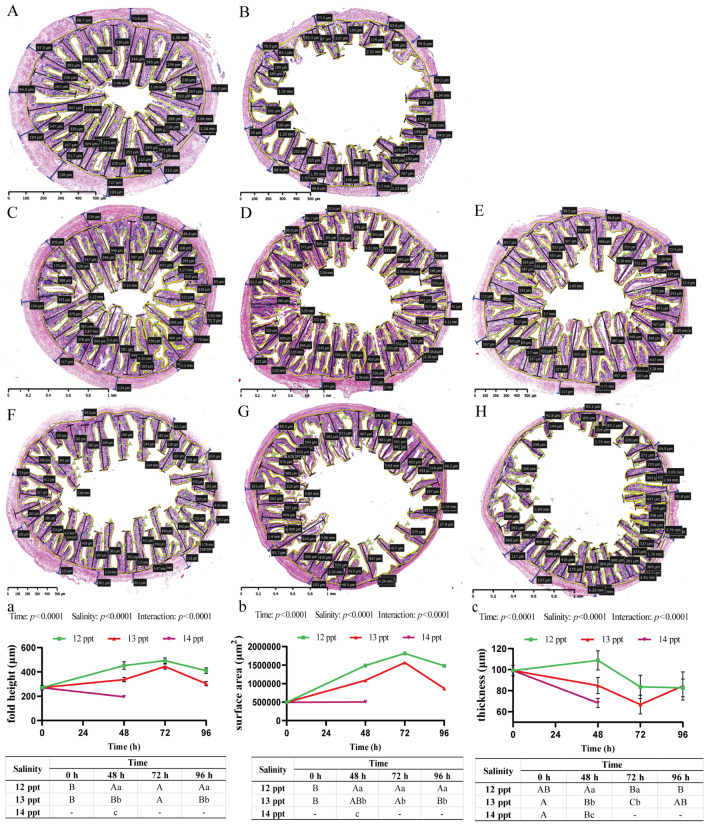
**Intestinal histopathological alterations and morphometric analysis in *Perca schrenkii* under acute salinity stress**. (**A**–**H**) Representative H&E-stained intestinal sections. (**A**) Control group. (**B**) 14 ppt (48 h). (**C**–**E**) 12 ppt: (**C**) 48 h; (**D**) 72 h; (**E**) 96 h. (**F**–**H**) 13 ppt: (**F**) 48 h; (**G**) 72 h; (**H**) 96 h. Markers: Black double-headed arrows = mucosal fold height; yellow brackets = mucosal fold range; blue double-headed arrows = muscularis thickness; green flags = goblet cells. Scale bar = 200 μm (all micrographs). (**a**–**c**) Intestinal morphometric parameter dynamics (*n* = 9, mean ± SE): (**a**) mucosal fold height, (**b**) total mucosal surface area, (**c**) muscularis thickness. Uppercase letters = inter-temporal differences (same salinity); lowercase letters = inter-salinity differences (same time) (Two-way ANOVA, Tukey’s test, *p* < 0.05). ANOVA results (time, salinity, interaction) are shown in each panel.

**Figure 4 biology-14-01775-f004:**
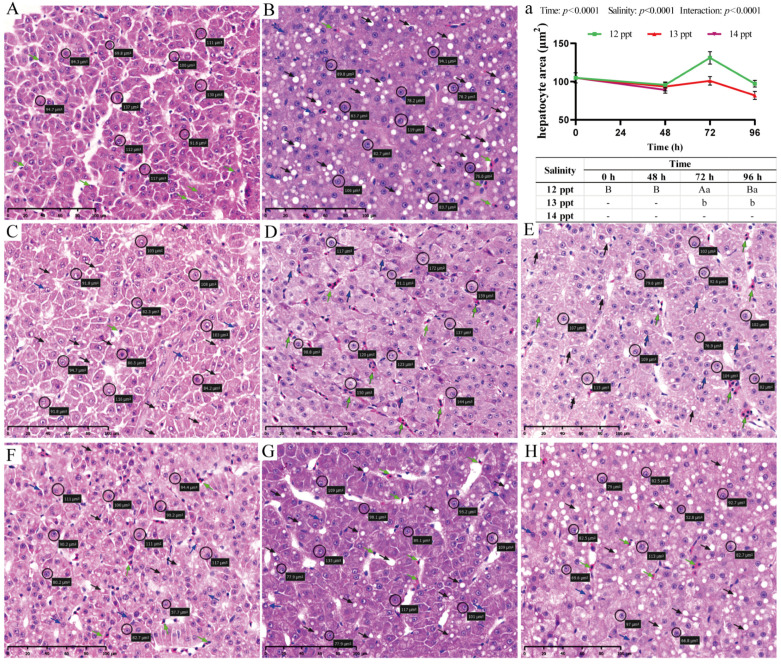
**Hepatic histopathological alterations and morphometric analysis in *Perca schrenkii* under acute salinity stress**. (**A**–**H**) Representative H&E-stained liver sections. (**A**) Control group. (**B**) 14 ppt (48 h). (**C**–**E**) 12 ppt: (**C**) 48 h; (**D**) 72 h; (**E**) 96 h. (**F**–**H**) 13 ppt: (**F**) 48 h; (**G**) 72 h; (**H**) 96 h. Markers: Circles = representative hepatocytes; blue arrows = vacuolation; green arrows = blood cell infiltration; black arrows = displaced nuclei. Scale bar = 50 μm (all micrographs). (**a**) Hepatocyte area dynamics (*n* = 9, mean ± SE) under different salinities. Uppercase letters = inter-temporal differences (same salinity); lowercase letters = inter-salinity differences (same time) (Two-way ANOVA, Tukey’s test, *p* < 0.05). ANOVA results (time, salinity, interaction) are shown in the panel.

**Figure 5 biology-14-01775-f005:**
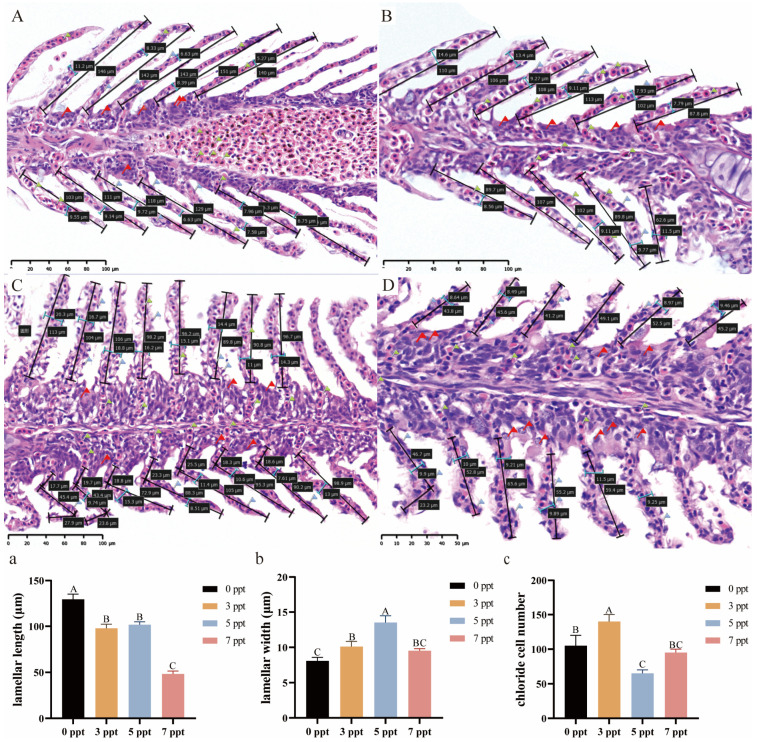
**Gill structural remodeling in *Perca schrenkii* after 60 day chronic salinity acclimation.** (**A**–**D**) Representative H&E-stained gill sections (60 day exposure): (**A**) Control; (**B**) 3 ppt; (**C**) 5 ppt; (**D**) 7 ppt. Markers: Black double-headed arrows = lamellar length quantified in (**a**); blue = lamellar width quantified in (**b**); red flags = chloride cells quantified in (**c**). Scale bar = 50 μm (all). (**a**–**c**) Gill parameter analysis (60 day chronic exposure, *n* = 9, mean ± SE): (**a**) lamellar length, (**b**) width, (**c**) chloride cell count. Different uppercase letters = inter-salinity differences (one-way ANOVA, Tukey’s test, *p* < 0.05).

**Figure 6 biology-14-01775-f006:**
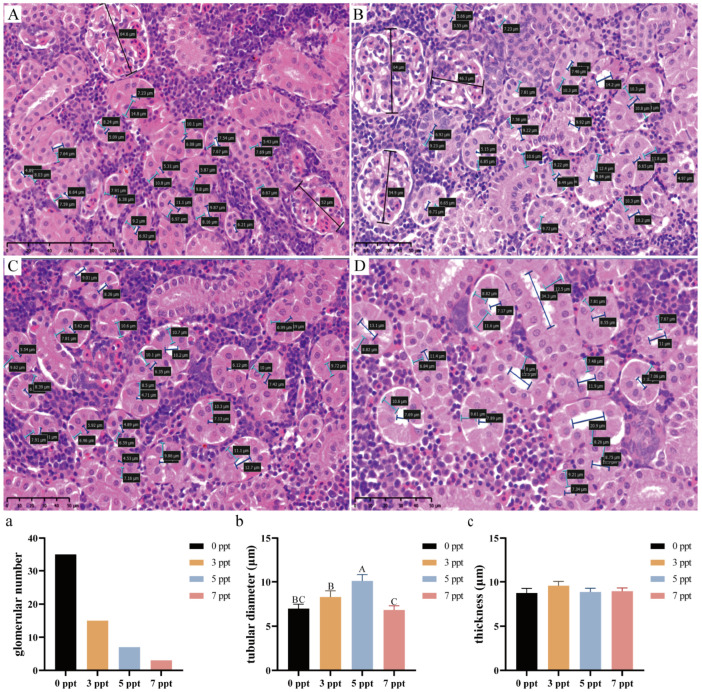
**Renal morphological changes in *Perca schrenkii* after 60 day chronic salinity exposure.** (**A**–**D**) Representative H&E-stained kidney sections (60 day exposure): (**A**) Control; (**B**) 3 ppt; (**C**) 5 ppt; (**D**) 7 ppt. Markers: Black double-headed arrows = glomerular longest diameter; blue = tubular diameter; turquoise = tubular epithelial thickness. Scale bar = 100 μm (all). (**a**–**c**) Renal parameter analysis (60 day chronic exposure, *n* = 9, mean ± SE): (**a**) glomerular number per field, (**b**) tubular diameter, (**c**) tubular epithelial thickness. Different uppercase letters = inter-salinity differences (one-way ANOVA, Tukey’s test, *p* < 0.05).

**Figure 7 biology-14-01775-f007:**
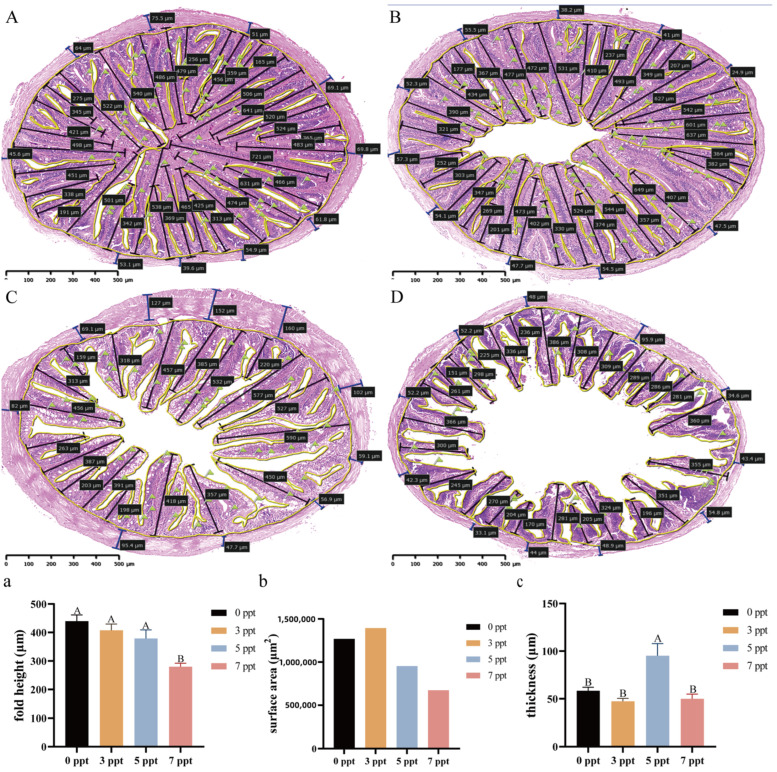
**Intestinal structural adaptation in *Perca schrenkii* after 60 day chronic salinity acclimation**. (**A**–**D**) Representative H&E-stained intestinal sections (60 day exposure): (**A**) Control; (**B**) 3 ppt; (**C**) 5 ppt; (**D**) 7 ppt. Markers: Black double-headed arrows = mucosal fold height; yellow brackets = mucosal fold range; blue = muscularis thickness; green flags = goblet cells. Scale bar = 200 μm (all). (**a**–**c**) Intestinal parameter analysis (60 day chronic exposure, *n* = 9, mean ± SE): (**a**) mucosal fold height, (**b**) total mucosal surface area, (**c**) muscularis thickness. Different uppercase letters = inter-salinity differences (one-way ANOVA, Tukey’s test, *p* < 0.05).

**Figure 8 biology-14-01775-f008:**
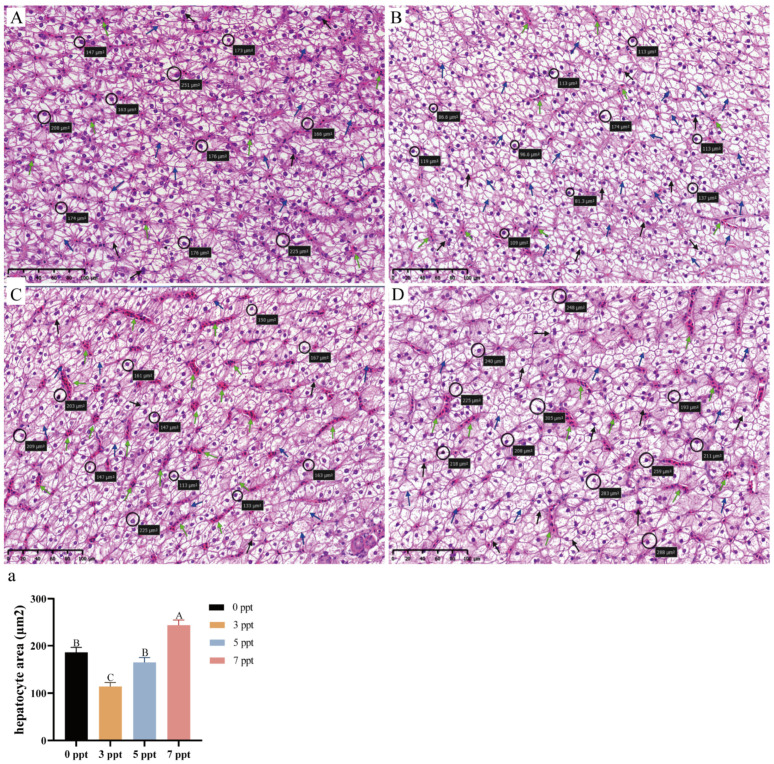
**Hepatic morphological changes in *Perca schrenkii* after 60 day chronic salinity exposure.** (**A**–**D**) Representative H&E-stained liver sections (60 day exposure): (**A**) Control; (**B**) 3 ppt; (**C**) 5 ppt; (**D**) 7 ppt. Markers: Circles = representative hepatocytes; blue arrows = vacuolation; green arrows = erythrocyte accumulation; black arrows = nuclear displacement. Scale bar = 50 μm (all). (**a**) Hepatocyte area analysis (60 day chronic exposure, *n* = 9, mean ± SE). Different uppercase letters = inter-salinity differences (one-way ANOVA, Tukey’s test, *p* < 0.05).

**Figure 9 biology-14-01775-f009:**
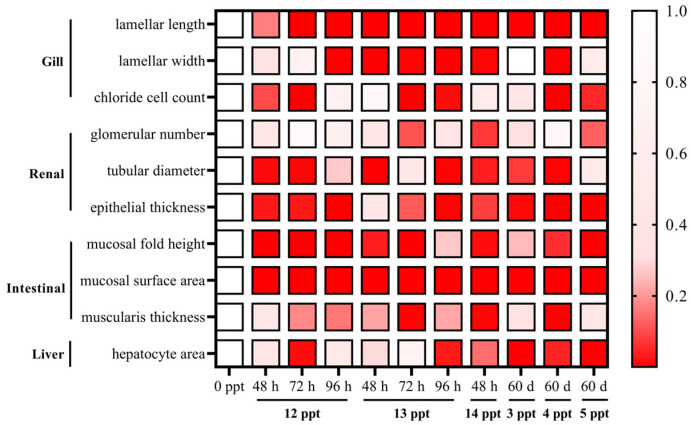
**Integrated heatmap summarizing the histomorphological responses across multiple organs to salinity stress (color scale: light red to dark red corresponds to *p*-values from 0.05 to <0.001; light gray indicates non-significant differences with *p* ≥ 0.05).** Heatmap shows difference significance (*p*-values: each group vs. control group). Rows = morphometric parameters from Figure 1, Figure 2, Figure 3, Figure 4, Figure 5, Figure 6, Figure 7 and Figure 8 (grouped by Gill/Kidney/Intestine/Liver); columns = acute (12–14 ppt) and chronic (3–7 ppt) salinity treatments. Color intensity indicates significance (darker = lower *p*-values, stronger effects); non-significant results (*p* ≥ 0.05) = light gray. It highlights organ-specific vulnerability and systemic adaptation across salinities.

**Table 1 biology-14-01775-t001:** Hierarchical Framework of Salinity Adaptation in *Perca schrenkii.*

Salinity Range	Classification	Key Histological Changes (Organs)	Physiological Meaning	Practical Recommendation
≤3 ppt	Safe Zone	No obvious alterations	Normal osmoregulation, long-term sustainable	Upper salinity limit recommended for culture
5 ppt	Acclimation Zone	Intestinal muscularis thickening, renal-tubule dilation, lamellar shortening	Active remodelling that enhances regulatory capacity	Optimal culture salinity, good health indicators
7 ppt	Tolerance Zone	Severe lamellar reduction, hepatocyte swelling, mucosal atrophy	Structural compensation with high energetic cost	Short-term culture only, close monitoring required
≥13 ppt	Lethal Zone	Lamellar fusion, chloride-cell necrosis, glomerular loss, hepatic degeneration	Multi-organ collapse, irreversible damage	Not suitable for culture, extremely high mortality risk

## Data Availability

All data generated or analyzed during this study are included in this published article.

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
