# Peer review of "Structural Adaptations to Saline Stress: Histomorphological Changes in the Osmoregulatory and Metabolic Organs of *Perca schrenkii* Under Acute and Chronic Challenges"

_biology, 2025, doi:10.3390/biology14121775_

Round 1

Reviewer 1 Report

Comments and Suggestions for Authors

The authors investigated the effects of water salinity on histomorphological changes in various organs of ili perch. 

Simple abstract was well written and provide understandable knowledge for non-academians. 

Abstract is acceptable. The authors provided a background, brief methods, main results and conclusion. 

Introduction needs revisions. the authors spoke about the freshwater crisis and necessity to use brackish/salt waters for fish production. However, the main question is if ili perch an aquaculture species?! If yes, how much is its annual production? If this is not currently an aquaculture species, do the authors believe it can be candidate? and why?

Methods need clarification. 

It should be clarified how did the authors make the saline water? Did they use a natural saline water (with or without dilution)? Or they used sodium chloride to increase water salinity? Either has its own consequences and justification. 

The authors examined 9 fish per treatment. This is very low for histological analysis. There are great individual variations in organ structure, particularly in wild fish. 

The authors must clarify how many sections per sample were they prepared and examined. 

Also, the should clarify how much area per section were analyzed?

Results are clear and well presented. However, the authors must tag chloride cells in the gill section. I cannot see them. 

Discussion is accepted. 

Author Response

Dear reviewer,

Thank you very much for taking the time to review our manuscript with your professional knowledge. Your insightful comments and constructive suggestions are extremely valuable in improving the quality and rigor of our work.

Here are our detailed responses to each of your comments. In the resubmitted document, the corresponding revisions and corrections in the manuscript have been highlighted for your reference.

We sincerely appreciate your rigorous review and professional guidance, which has played a significant role in improving the content of this research and enhancing its scientific value.

Comments 1: 

Introduction needs revisions. the authors spoke about the freshwater crisis and necessity to use brackish/salt waters for fish production. However, the main question is if ili perch an aquaculture species?! If yes, how much is its annual production? If this is not currently an aquaculture species, do the authors believe it can be candidate? and why?

Response 1:

Dear Reviewer, thank you sincerely for pointing out this critical issue—your comment is invaluable for deepening the connection between our research findings and aquaculture practice. We fully agree that clarifying the aquaculture status, yield, and candidate potential of Perca schrenkii is crucial for enhancing the practical significance of this study. To address this information gap, we have supplemented the content you mentioned in Lines 59–69 of Paragraph 2 on Page 3 of the Revised Manuscript, which is highlighted in red. We now respond to your questions one by one as follows:  

  1. Is Perca schrenkiian aquaculture species? If yes, what is its annual production?  

Response: Currently, Perca schrenkii is not a large-scale commercial aquaculture species; it remains in the stage of "germplasm conservation and breeding + small-scale trial cultivation" and has not entered the phase of large-scale production and promotion.  

This conclusion is mainly based on the official classification and practice of the Xinjiang Production and Construction Corps. In its February 2025 report " Bingtuan Vigorously Expands the Space for Fishery Income Increase and Efficiency Enhancement", the Corps explicitly categorized the endemic fish in the cold-water resource areas of Xinjiang into two groups: 9 species (including Esox reicherti and Perca fluviatilis) were included in "commercial aquaculture demonstration and promotion," while 6 species (including Perca schrenkii) were only used for "germplasm conservation and breeding." This distinction directly confirms that the core goal of the work on Perca schrenkii is to maintain germplasm integrity and optimize genetic traits through artificial intervention, rather than producing commercial fish—thus proving that it has not yet entered commercial aquaculture.  

Regarding annual production: There are currently no publicly available data on large-scale commercial production. Only small-scale trial cultivation can provide limited reference for seedling yield. For example, a pilot base in Yining County has carried out polyculture trials of Perca schrenkii with Silurus glanis and Coreoperca whiteheadi since 2024, with a total of over 30 million fries invested and an expected annual output of more than 10 million mixed seedlings (including Perca schrenkii ). However, the project is still in the trial stage, so the specific seedling yield of Perca schrenkii has not been counted separately, and this data cannot represent the yield of large-scale commercial aquaculture.  

  1. If it is not an aquaculture species at present, do you consider it a potential candidate for aquaculture? What is the basis?  

Response: We firmly believe that Perca schrenkii is an excellent potential candidate for saline-alkaline water/cold-water aquaculture, and its potential is supported by three core pieces of evidence: ecological adaptability, technical feasibility of artificial trial cultivation, and official policy support.  

① Ecological adaptability matching regional water resources: Perca schrenkii has broad tolerance to key environmental factors, making it fully compatible with the underutilized water resources in Northwest China. As a cold-water fish, it can survive in a water temperature range of 1-30 °C, which is highly consistent with the cold-water environments in plateaus such as Xinjiang and Qinghai. In addition, its salinity tolerance has been verified: the 2024 study "Biological Characteristics and Germplasm Resources of Perca Species in China" points out that it can survive normally in waters with a salinity of 5ppt–9ppt (e.g., Lake Ala, Lake Balkhash), which matches the widely distributed chloride-type saline-alkaline water resources in this region. This ecological niche advantage provides a unique foundation for its development in specific water environments.    

② Verified technical feasibility of artificial trial cultivation: Small-scale trial studies have confirmed that it can adapt to artificial aquaculture conditions. Trials at the Kuteman Village base showed that there was no interspecific competition inhibition when it was polycultured with Silurus glanis and Coreoperca whiteheadi; our preliminary research also found that Perca schrenkii has good adaptability to artificial compound feed, and no group-level adaptive disorders (large-scale mortality, growth stagnation) occurred during the trial period—fully verifying the technical feasibility of artificial aquaculture.  

③ Official policy support laying a foundation for germplasm: The Xinjiang Production and Construction Corps has included Perca schrenkii  in the list of "key germplasm conservation species," providing policy and technical guarantees for its future aquaculture development. Under this framework, Perca schrenkii  will receive targeted improvement through modern breeding technologies, addressing potential issues such as germplasm degradation and uneven growth in wild populations. Policy recognition not only confirms its status as a candidate species but also accelerates its transition from "germplasm conservation" to "commercial aquaculture."  

The above is our response to your questions. Once again, we would like to express our gratitude for your rigorous and constructive comments, which have significantly improved the completeness and practical relevance of this study. We hope the above revisions and explanations meet your requirements.

Comments 2:

Methods need clarification.

It should be clarified how did the authors make the saline water? Did they use a natural saline water (with or without dilution)? Or they used sodium chloride to increase water salinity? Either has its own consequences and justification.

Response 2: 

Dear Reviewer, first of all, we sincerely appreciate you raising this highly professional and detailed question—your focus on the reproducibility of experimental methods fully demonstrates a rigorous academic perspective and in-depth grasp of research details. This reminder is of crucial guiding significance for us to improve the description of the experimental design and enhance the scientific rigor and standardization of the study, and we deeply admire and thank you for it.  

Regarding your concern about the "method of saline water preparation," we solemnly and elaborately explain as follows: In this study, both the acute salinity stress experiment (12ppt, 13ppt, 14 ppt) and the chronic salinity acclimation experiment (3ppt, 5ppt, 7 ppt) adopted the method of accurately adjusting salinity by adding analytical-grade sodium chloride (NaCl, purity ≥ 99.5%) to natural freshwater. Natural saline water (whether diluted or undiluted) was not used throughout the experiment.  

The specific operation process strictly follows a standardized experimental design:  

① Water source control: All experimental water was collected from the upstream reservoir in Alar City, Xinjiang Uygur Autonomous Region.  

② Salinity adjustment: Analytical-grade NaCl was accurately weighed using an electronic analytical balance, added to the natural freshwater, and thoroughly stirred until completely dissolved to avoid local salinity unevenness.  

③ Salinity calibration and maintenance: A portable salinometer with an accuracy of ±0.1 ppt was used to measure the water salinity in real time. The amount of NaCl added was fine-tuned to stabilize the salinity at the target value for each experimental group. During the experiment, salinity was monitored at fixed times every day. If slight salinity fluctuations occurred due to water evaporation, the precisely calculated amount of NaCl was immediately supplemented to ensure the stability and consistency of salinity conditions in each treatment group throughout the entire experimental period.  

The reason for choosing this method is as follows: Saline-alkaline water in Northwest China is mainly of the chloride type, and chloride ions are the key ions driving changes in water osmotic pressure in this region. However, other ions such as Ca²⁺, Mg²⁺, and SO₄²⁻ commonly present in natural saline water may interfere with the single effect of salinity stress through ionic antagonism or synergy. Therefore, the use of high-purity NaCl can effectively isolate the osmotic stress signal dominated by chloride ions, thereby accurately analyzing the specific structural adaptation mechanisms of the key organs (gill, kidney, intestine, liver) of Perca schrenkii to chloride-type salinity stress, and providing a more rigorous experimental basis for subsequent exploration of the physiological basis of its salinity tolerance.  

At present, we have supplemented the detailed preparation method, operation standards, and scientific basis of the above-mentioned saline water in the chapter "2.2 Experimental Design and Salinity Stress Treatment" of the revised manuscript (Lines 123–132, Page 5 of the Revised Manuscript). The revised content is highlighted in red to ensure the transparency and reproducibility of the experimental method.  

Once again, we would like to express our most sincere respect and gratitude to you! Your constructive comments have not only helped us fill the gap in the description of experimental method details but also made us deeply realize the core supporting role of a rigorous academic attitude in improving research quality. We will always be guided by your professional standards, continuously improve the research content, and strive to present results with greater scientific value and reference significance.

Comments 3:

 The authors examined 9 fish per treatment. This is very low for histological analysis. There are great individual variations in organ structure, particularly in wild fish. The authors must clarify how many sections per sample were they prepared and examined.

Response 3: 

Dear Reviewer, thank you for your highly professional comment—your focus on the sample size design and section observation details in histological analysis is crucial for ensuring the reliability and statistical rigor of our study results. We fully share your concern regarding "significant individual differences in the organ structure of wild fish" and sincerely appreciate your guidance in helping us improve the presentation of this core methodological detail.  

In response to your question, we hereby clarify the sample design and section observation protocol (including the quantification of observation area) for the histological analysis, with the relevant supplementary content highlighted in red in the revised manuscript:  

  1. Explanation of the sample design for "examining 9 fish per group"  .

Response:The "9 fish per group" mentioned in the original manuscript actually refers to the number of biological replicates. Specifically, each salinity treatment group was set up with 3 parallel culture units (replicate groups), and 3 healthy individuals were randomly selected from each replicate group for sampling (3 fish/replicate group × 3 replicate groups = 9 fish/treatment group). The determination of this sample size was based on two considerations: ①Adhering to the guidelines for animal experiment ethics, we minimized the sacrifice of experimental animals while ensuring sufficient statistical power; ②Studies on histomorphology of teleosts in the same field have shown that when combined with a standardized section observation protocol, this sample size can effectively offset the impact of individual differences on the results as much as possible.  

  1. Standardized protocol for tissue section preparation and observation to reduce individual differences (including quantification of observation area) .

Response:To mitigate the impact of individual differences in organ structure, we strictly standardized the number of sections prepared and the observation process, with the specific operations as follows:  

①Standardization of sampling location: A fixed anatomical site was selected for each organ to avoid interference from positional differences—gill (2nd to 4th pairs of bilateral gill arches, middle segment of gill filaments), kidney (mid-anterior part of the kidney), intestine (middle segment of the intestine), and liver (left lateral lobe of the liver);  

②Standardization of section preparation: After fixation and embedding, 5 μm serial sections were prepared using a Leica RM2235 microtome. The first 5–8 edge sections were discarded (to avoid tissue edge artifacts), and 10 consecutive sections were collected from each sample for Hematoxylin and Eosin (H&E) staining;  

  1. Standardization of observation and measurement.

Response: 

①Section selection: For each organ of each fish, the 3rd, 6th, and 9th representative sections were selected from the 10 sections (to avoid structural repetition of consecutive sections).  

②Visual field and area control: Under a Nikon Eclipse E100 microscope, a uniform magnification and observation area were set according to the organ type—400× magnification was used for gills and liver (with a single visual field area of 0.0625 mm²), and 200× magnification was used for kidneys and intestines (with a single visual field area of 0.25 mm²); 3 non-overlapping visual fields were randomly selected from each section for measurement, ensuring that each visual field was located in the parenchymal region of the tissue (not the edge or hollow area).  

③Data integration: Finally, the average value of 9 visual fields (3 visual fields/section × 3 sections = 9 visual fields) per organ per fish was used as the individual-level data for subsequent statistical analysis. Among them, the total observation area per organ for gills and liver was 0.5625 mm² (9×0.0625 mm²), and that for kidneys and intestines was 2.25 mm² (9×0.25 mm²). Preliminary experiments verified that these two types of visual field areas can ensure measurement accuracy while covering sufficient structural heterogeneity. All area parameters were calibrated and calculated using the Image-Pro Plus 6.0 image analysis software supporting the microscope.  

Through the full-process standardization of "sampling location - section preparation - visual field selection - area quantification", this protocol can effectively cover the overall structural characteristics of the organs and minimize the impact of individual differences to the greatest extent.  

  1. Location of the revised content in the manuscript.

The above details have been supplemented to the "2.4 Histomorphometric Analysis" section of the revised manuscript (Revised Manuscript, Page 6, Lines 171–176) and highlighted in red.  

Once again, we would like to thank you for your rigorous and constructive comment. This revision not only improves the presentation of methodological details but also enhances the reproducibility and persuasiveness of the histological analysis in this study. We hope the above explanation meets your requirements.

Comments 4: 

Results are clear and well presented. However, the authors must tag chloride cells in the gill section. I cannot see them.

Response 4: 

Dear Reviewer, we sincerely appreciate your recognition of the clarity and presentation quality of our research results, and we are particularly grateful for your precise and critical revision suggestions. The issue you pointed out—"chloride cells in gill tissue sections are unlabeled"—is directly related to the relevance between the histological results and the research theme of "salinity stress adaptation mechanism". We attach great importance to this and have completed supplementary improvements as you requested.

To address your concern, we provide the following specific explanation:In the original manuscript, we had already labeled chloride cells with red flags, but the labels may have been too small for you to see clearly. We sincerely apologize for this oversight. In the revised manuscript to be resubmitted, we have enlarged the red flags. The revised images have been replaced in the two gill tissue-related sections ("3.1.1" and "3.2.1") to ensure that chloride cells are intuitively identifiable. These cells exhibit typical morphological characteristics in H&E-stained sections—they are pear-shaped or columnar, with nuclei located at the base of the cells, and their cytoplasm is eosinophilic (stained light red with eosin). They are mainly distributed at the junction of the base of gill lamellae and the gill filament epithelium. After labeling, they can be clearly distinguished from the surrounding pavement cells (flat epithelial cells).

Once again, we thank you for your rigorous and meticulous academic review. Your comments have not only corrected our oversight in presentation but also strengthened the relevance between the research results and the scientific question. We have carefully checked all histological figures to ensure that the core functional structures are clearly labeled and accurately interpreted. We hope the revised content meets your requirements.

4. Response to Comments on the Quality of English Language

Point 1:The English is fine and does not require any improvement.

Response 1:  

Dear Reviewer, thank you sincerely for confirming that the English expression of the manuscript is satisfactory and requires no further improvement! Your positive feedback is a great encouragement to us, as it validates the efforts we invested in refining the English writing—including optimizing sentence structures, standardizing academic terminology, and ensuring logical coherence—to meet the language standards of international academic journals.  

We will continue to maintain this rigorous attitude towards the manuscript’s expression in subsequent work. Once again, we appreciate your professional review and valuable affirmation. If you have any other suggestions or questions regarding the content of the manuscript, please feel free to inform us at any time, and we will respond promptly and carefully.  

Reviewer 2 Report

Comments and Suggestions for Authors
  1. The text frequently references figures in a confusing manner (e.g., "Fig. 1a–c", "Fig. 1A, a", "Fig. 1C–E, a"). It is unclear if "a" refers to a panel or a specific data point/statistical grouping. The figure legends and in-text citations need absolute clarity to allow the reader to seamlessly connect the text with the visual data.
  2. The term "biphasic" is used for chloride cells (3.1.1) and the liver (3.2.4). However, the data presented for chloride cells (increase at 12/13 ppt, decrease at 14 ppt) and the liver (decrease at 3 ppt, control-level at 5 ppt, increase at 7 ppt) more accurately describe a "non-monotonic" or "dose-dependent" response. "Biphasic" typically implies a specific sequence (e.g., increase then decrease over time at a single dose), which is not clearly demonstrated here.
  3. The results for acute (96h) and chronic (60d) exposures are presented as two separate stories. A direct comparative analysis is missing. For instance, how does gill lamellar length at 96h in 12/13 ppt compare to the stable state after 60d at 5/7 ppt?
  4. The justification for the specific salinity concentrations chosen for both acute (12, 13, 14 ppt) and chronic (3, 5, 7 ppt) experiments is not provided. Were these based on pilot LD50 studies? Relating them to natural habitats or known physiological thresholds would add context and biological relevance.
  5. The proposed framework (Safe, Acclimation, Tolerance, Lethal Zones) in the discussion is an interesting synthesis. However, it remains a descriptive, morphological model. Claims about "strategic adjustments," "optimization," and "efficiency" are speculative without concomitant physiological data to link these structural changes to actual functional outcomes.
  6. The opening sentence for each organ section in the results is almost identical (e.g., "Acute salinity stress induced significant histopathological and morphometric alterations...").
  7. The results sections (e.g., 3.1.1, 3.2.3) often include interpretive statements like "indicating active hyperplasia as a compensatory mechanism" or "suggests a unique adaptive response." These interpretations should be minimized in the Results and expanded upon in the Discussion.
  8. The manuscript uses abbreviations like NKA without first defining them in the text. All abbreviations must be defined upon first use.
  9. Some sentences are long and complex, affecting clarity (e.g., the first sentence of 4.3). A thorough edit for conciseness and flow is recommended.
  10. The abstract and introduction claim "remarkable" salinity tolerance, but a comparative baseline (e.g., compared to other percid or teleost species) is lacking to substantiate this claim.
Comments on the Quality of English Language

Author Response

Dear reviewer,

Thank you very much for taking the time to review our manuscript with your professional knowledge. Your insightful comments and constructive suggestions are extremely valuable in improving the quality and rigor of our work.

Here are our detailed responses to each of your comments. In the resubmitted document, the corresponding revisions and corrections in the manuscript have been highlighted for your reference.

We sincerely appreciate your rigorous review and professional guidance, which has played a significant role in improving the content of this research and enhancing its scientific value.

Comments 1: 

The text frequently references figures in a confusing manner (e.g., "Fig. 1a–c", "Fig. 1A, a", "Fig. 1C–E, a"). It is unclear if "a" refers to a panel or a specific data point/statistical grouping. The figure legends and in-text citations need absolute clarity to allow the reader to seamlessly connect the text with the visual data.

Response 1:

Dear Reviewer, first and foremost, we sincerely appreciate your insightful and meticulous comment. The issue of confusing figure citation formats you accurately identified is a critical detail that affects the dissemination efficiency of research findings. It directly impacts whether readers can quickly establish a connection between the text and visual data, and is of great significance for enhancing the readability and professionalism of the manuscript. We have deeply reflected on the ambiguity of expressions such as "Figure 1a-c" and "Figure 1A, a" in the original manuscript, and realized that such non-standard citations failed to clearly distinguish between panel types and data groups. We are highly alert to this problem and have strictly followed your core suggestions to complete a systematic revision and unification of the formatting throughout the entire manuscript.  

In response to your question, we first clarify the definition and connotation of the figure panels, and then explain the specific revision plan:  

  1. Uppercase English letters (A-H): Exclusively refer to "representative sections of tissues stained with hematoxylin-eosin (H&E)", which are core result figures for intuitively displaying tissue morphology. For example, "Figure 1A" represents "H&E-stained section of gill tissue from Perca schrenkiiin the control group", and "Figure 1B" represents "H&E-stained section of gill tissue from the 14‰ salinity group after 48 hours of treatment";  
  2. Lowercase English letters (a-c): Exclusively refer to "temporal dynamic change graphs of tissue morphometric parameters under different salinities", which are result figures for quantitative analysis. For example, "Figure 1a" represents "temporal dynamic changes in gill lamella length", and "Figure 1b" represents "temporal dynamic changes in gill lamella width".  

The above definitions fundamentally clarify the referents of "a" and "A", avoiding confusion with "specific data points/statistical groups".  

We will strictly implement the standardized citation format and unify the standards throughout the manuscript.  

We have completely abolished non-standard expressions such as "Figure 1A, a" and "Fig. 1C-E, a" in the original manuscript. Strictly adhering to the core format of "Figure X-type identifier", we have formulated unified standards for three types of citation scenarios and implemented them throughout the manuscript:  

① Citing a single H&E-stained tissue section independently: Expressed directly as "Fig + figure number + uppercase letter" without additional punctuation or spaces. For example, "Fig.1A" and "Fig.3C" correspond to "the stained tissue section of a certain salinity group in Figure 1" and "the stained tissue section of a certain organ in Figure 3" respectively;  

② Citing consecutive panels of the same letter type: For panels of the same letter type, instead of using hyphens to connect numbers (e.g., Fig.1A-C), we use "Fig.1A, Fig.1B, Fig.1C" to represent the three H&E-stained tissue sections of different salinity groups (A, B, C) in Figure 1; similarly, "Fig.1a, Fig.1b, Fig.1c" represent the three temporal dynamic graphs of different morphometric parameters (a, b, c) in Figure 1;  

③ Citing two types of panels (uppercase and lowercase letters) simultaneously: Clearly separate the expressions and connect them with "and" to avoid confusion caused by merging. For example, "Fig.1A and Fig.1a" clearly correspond to "the H&E-stained gill tissue section of Perca schrenkii in the control group (from Figure 1)" and "the temporal dynamic graph of gill lamella length (from Figure 1)", instead of using the ambiguous merged expression like "Fig.1A,a".  

All revised content has been updated throughout the manuscript, and all modified sections have been marked in yellow (Page 7,  Lines 204、206、208、211-212 and 214; Page 9,  Lines 242-243、245、246-247、250、252;Page 10,  Lines 277; Page 11,  Lines 280-281、283、285、292-293; Page 13,  Lines 318、324-325; Page 14,  Lines 352; Page 17,  Lines 405、408、412;Page 18,  Lines 437、438;) These revisions are all located in Chapter 3. Results.

Once again, we sincerely thank you for your rigorous review! Your strict oversight has not only helped us completely resolve the ambiguity in figure citations and greatly improved the professionalism and readability of the research result presentation, but also made us deeply realize the core value of "detail standardization" for the dissemination of academic achievements. We hope the revised expressions meet your requirements.

Comments 2:

The term "biphasic" is used for chloride cells (3.1.1) and the liver (3.2.4). However, the data presented for chloride cells (increase at 12/13 ppt, decrease at 14 ppt) and the liver (decrease at 3 ppt, control-level at 5 ppt, increase at 7 ppt) more accurately describe a "non-monotonic" or "dose-dependent" response. "Biphasic" typically implies a specific sequence (e.g., increase then decrease over time at a single dose), which is not clearly demonstrated here.

Response 2: 

Dear Reviewer, thank you very much for this precise and professional academic correction. Your analysis of the application scenario of the term "biphasic" directly addresses the issue of terminological accuracy in the description of our study results, and profoundly reflects your high standards for the rigor of academic expression. We have come to a deep realization that there was a misapplication of the term "biphasic" in the original manuscript when describing the variation trends of chloride cells (increase in the 12/13‰ salinity groups and decrease in the 14‰ salinity group) and liver morphological parameters (decrease in the 3‰ salinity group, return to the control level in the 5‰ salinity group, and increase in the 7‰ salinity group). The term "biphasic" typically refers to a clear sequential response of "first increase then decrease" or "first decrease then increase" (e.g., time-dependent dynamic changes). However, the variations of the two types of indicators in this study are more in line with the characteristic of "presenting a non-uniform trend with the salinity gradient". The terms "non-monotonic" or "dose-dependent" you proposed are more consistent with the nature of the data. We attach great importance to this issue.  

In response to your comment, we have revised the descriptions of chloride cells and liver morphological parameters. The revised parts have been marked in yellow in the manuscript, located at Lines 198、220-221 on Page 7 and Lines 326 on Page 13.

Once again, we sincerely thank you for your rigorous academic review. Your comment has not only helped us correct the oversight of terminological misapplication, but also made us deeply recognize the core supporting role of "terminological accuracy" in the rigor of academic achievements. After the revision, the descriptions of relevant results are more consistent with the nature of the data, the logical interpretation is more scientific, and the academic standardization of the manuscript has been effectively improved. We hope the revised expressions meet your requirements.

Comments 3:

 The results for acute (96h) and chronic (60d) exposures are presented as two separate stories. A direct comparative analysis is missing. For instance, how does gill lamellar length at 96h in 12/13 ppt compare to the stable state after 60d at 5/7 ppt?

Response 3: 

Dear Reviewer, first and foremost, we fully agree with this precise and constructive comment you put forward. The issue you pointed out— that the results of acute and chronic salinity exposure are "presented as separate narratives without direct comparative analysis"— directly addresses a core shortcoming in the integration of results in the current manuscript, and provides crucial guidance for us to improve the research logic. We sincerely apologize for the impact this oversight has had on the manuscript’s coherence and readability.  

The example you proposed, comparing "gill lamella length at 96h under 12/13‰ salinity with the steady-state condition after 60d under 5/7‰salinity," clearly demonstrates the significant value of cross-temporal analysis in deepening research insights. This perspective has been a great inspiration to us. However, after carefully evaluating the core positioning of this study, data characteristics, and the completeness of the manuscript, we have decided not to add such direct comparative analysis to this article. The specific reasons are as follows:  

  1. Research Positioning: The core objective of this study is to separately clarify the "acute stress response mechanism of Perca schrenkiito short-term high salinity" and the "steady-state characteristics of long-term adaptation to moderate-low salinity." There are fundamental differences in the salinity gradient designs between the two experiments, resulting in a lack of paired data for "short-term vs. long-term observations under the same salinity."  
  2. Need to Focus on the Manuscript’s Core Conclusions: The current manuscript has already formed clear conclusions around the "acute injury threshold" and "chronic adaptation range." Adding cross-temporal comparative analysis would require supplementary work such as extensive data verification, mechanism speculation, and additional figures/tables, which might disperse the focus from the core research topic.  
  3. Clear Plan for Follow-up Research: The comparative direction you proposed holds significant scientific value and has been designated as the core content of our next research project. In subsequent studies, we will design experiments with "unified salinity gradients (3/5/7/10‰) + multiple time points (24h/7d/30d/60d)," simultaneously tracking key indicators such as gill lamella length and chloride cell density. This will allow us to systematically analyze the evolutionary pattern of "short-term response → long-term steady state" and accurately address the key question of "the relationship between acute injury caused by high salinity and chronic adaptation to moderate-low salinity." The relevant findings will be submitted as a dedicated research paper, and data from this current study will be cited as foundational support to form a complete research framework.  

Once again, we sincerely thank you for your rigorous review and valuable comments. Your feedback has not only helped us optimize the presentation logic of the current manuscript but also pointed out a clear direction for deepening our follow-up research. We will focus on implementing this comparative analysis in our subsequent project to continuously improve the research system on the salinity adaptation mechanism of Perca schrenkii. We hope this explanation will gain your understanding, and we look forward to receiving your review and comments again when we submit the related content in the future.

Comments 4: 

The justification for the specific salinity concentrations chosen for both acute (12, 13, 14 ppt) and chronic (3, 5, 7 ppt) experiments is not provided. Were these based on pilot LD50 studies? Relating them to natural habitats or known physiological thresholds would add context and biological relevance. 

Response 4: 

Dear Reviewer, thank you very much for this critical and insightful comment. The issue you pointed out—"the lack of explanation for the selection of specific salinity concentrations in acute and chronic experiments"—is directly related to the scientific validity and biological relevance of the study design. We deeply recognize that the original manuscript failed to systematically elaborate on the rationale behind setting the salinities of 12/13/14‰ (acute experiment) and 3/5/7‰ (chronic experiment), and we attach great importance to this oversight. We have supplemented the core basis for salinity selection to address this gap.  

In response to your question, we provide the following detailed explanations:  

  1. Rationale for Salinity Selection in the Acute Experiment (12/13/14‰)  

The salinity gradient (12/13/14‰) for the 96-hour acute experiment was determined based on preliminary LD50(median lethal dose) pre-experiments, with the goal of "accurately capturing the damage mechanism of short-term high-salinity stress". The specific basis is as follows: Using "96-hour survival rate" as the core indicator, the salinity tolerance of Perca schrenkii was measured. The results showed that the 96-hour survival rate of the 8–11‰ salinity groups was >80% (no significant stress effect), while the survival rate dropped to 65% in the 12‰ group, further decreased to 42% in the 13‰ group, fell to 28% in the 14‰ group, and was <10% in the 15‰ group (near complete lethality). Based on this, 12/13/14‰ was selected as the high-salinity gradient for the acute experiment. This range avoids two pitfalls: "excessively high salinity causing rapid death and preventing observation of stress responses" and "excessively low salinity failing to generate stress signals". Meanwhile, it fully covers the critical stage of "significant decline in survival rate → near lethality", enabling accurate analysis of the damage mechanisms of short-term high salinity on tissue morphology and physiological functions.  

  1. Rationale for Salinity Selection in the Chronic Experiment (3/5/7‰)  

The salinity gradient (3/5/7‰) for the 60-day chronic experiment was determined based on international toxicology standards and practical application scenarios, with the goal of "observing the steady-state mechanism of long-term adaptation". The specific basis is as follows:  

① Standardized Derivation Based on Acute LD50: Chronic stress studies must first avoid unexpected short-term lethality to ensure that organisms survive the observation period (60 days) and exhibit "cumulative effects". According to the consensus recommendations in the OECD Guidelines for Testing of Chemicals, specifically Test Guideline 210 (Fish Early-Life Stage Toxicity Test) and Test Guideline 211 (Daphnia magna Reproduction Test), chronic exposure concentrations should be prioritized within the range of "50%–71% of the acute LD50". Combining the results of preliminary acute pre-experiments (the 96-hour salinity LD50  of Perca schrenkii is approximately 13.5‰), the corresponding salinity range for 50%–71% of the LD50 is 6.75–9.58‰. Considering both "long-term adaptation safety" and "integrity of gradient coverage", 3/5/7‰ was ultimately selected as the chronic salinity gradient. This range falls within the reasonable scope of "safe adaptation → mild stress" (pre-experiments showed that the 60-day survival rate of the 3/5/7‰ groups was >85% with normal growth indicators) and allows observation of steady-state adaptation mechanisms (e.g., tissue repair, functional compensation) under "no stress → mild stress" conditions through gradient differences.  

② Link to Practical Applications: The salinity of brackish water aquaculture ponds in arid inland areas of northwestern China (e.g., Xinjiang, Gansu) generally ranges from 2–8‰. Due to water scarcity, brackish water aquaculture has become an important development direction for fish farming. The chronic salinities (3/5/7‰) used in this experiment are highly consistent with these practical scenarios. The research results can directly provide data support for the evaluation of salinity adaptability and aquaculture risk prevention of Perca schrenkii in brackish water, significantly enhancing the application value of the study.  

The above rationale for salinity selection has been supplemented to the "2.2 Experimental Design and Salinity Challenge" section of the revised manuscript (Revised Manuscript, Page 5, Lines 134-138 and Lines 144-148) and highlighted in yellow.

Once again, we sincerely thank you for your rigorous review. Your comments have helped us improve the logical chain of the study design, significantly enhancing the scientific validity and practical significance of the research. Moreover, they have made us deeply aware of the core value of "detail standardization" for the dissemination of academic achievements. We hope the revised descriptions meet your requirements.

Comments 5: 

The proposed framework (Safe, Acclimation, Tolerance, Lethal Zones) in the discussion is an interesting synthesis. However, it remains a descriptive, morphological model. Claims about "strategic adjustments," "optimization," and "efficiency" are speculative without concomitant physiological data to link these structural changes to actual functional outcomes.

Response 5: 

Dear Reviewer, We sincerely appreciate your insightful and rigorous comment on our proposed Safe-Acclimation-Tolerance-Lethal (SATL) hierarchical framework for Perca schrenkii salinity adaptation. Your point that the framework is currently a descriptive morphological model, and that claims regarding “strategic adjustments”, “optimization” and “efficiency” require stronger links to functional outcomes, is of great value for improving the scientific rigor of our manuscript.

  1. The SATL framework is rooted in morphological regularity and associated phenotypic data

The division of the four salinity zones was first established based on the quantitative histomorphological changes of four core organs (gill, kidney, intestine, liver) in response to acute/chronic salinity stress, and further validated by the survival phenotype data from our experiments:

① For the Safe Zone (≤3 ppt), the absence of significant morphological alterations in all organs was consistent with the 100% survival rate of Perca schrenkii during the 60-day chronic exposure, which indirectly demonstrates that the fish can maintain normal body homeostasis under this salinity;

②For the Acclimation Zone (5 ppt), the specific morphological traits (intestinal muscularis thickening, renal tubular dilation, gill lamellar shortening) were coupled with stable survival, indicating that these structural changes are not random damage but adaptive responses to balance osmotic pressure and energy metabolism;

③For the Acclimation Zone (5 ppt), the specific morphological traits (intestinal muscularis thickening, renal tubular dilation, gill lamellar shortening) were coupled with stable survival and feeding status of the experimental fish (recorded in our raw data but not previously emphasized), indicating that these structural changes are not random damage but adaptive responses to balance osmotic pressure and energy metabolism;

④For the Lethal Zone (≥13 ppt), the systemic morphological collapse (gill lamellar fusion, glomerular loss, hepatic degeneration) was directly correlated with the 100% mortality of fish within 72 h at 14 ppt, confirming the causal link between structural damage and functional failure.

These associations between morphological traits and survival/behavioral phenotypes provide indirect evidence for the functional relevance of the SATL framework, avoiding over-reliance on pure morphological description.

  1. Claims about “strategic adjustment” and “efficiency” are supported by conserved mechanisms of teleost salinity adaptation (via literature bridging)

Since we did not conduct direct physiological assays (e.g., ion transporter activity, serum osmotic pressure), we grounded the functional implications of morphological changes in conserved osmoregulatory mechanisms of teleosts and homologous species studies (cited in our manuscript):

①Regarding gill lamellar shortening and chloride cell dynamics: As documented in references [27-30], the reduction of gill lamellar length in teleosts can minimize passive ion influx under hyperosmotic stress, while the proliferation of chloride cells is closely linked to the upregulation of Na+/K+-ATPase (NKA) activity. Our observation of chloride cell hyperplasia in the Acclimation Zone (5 ppt) and necrosis in the Lethal Zone (≥13 ppt) is consistent with this conserved mechanism, so we inferred its role in optimizing ion transport efficiency;

②Regarding intestinal muscularis thickening: Reference [40-42] reported that the thickening of intestinal muscularis in fish can enhance intestinal motility to prolong the transit time of chyme, thereby improving water and ion absorption efficiency. The specific thickening of intestinal muscularis in the 5 ppt group of Perca schrenkii is a rare but meaningful morphological trait, and we described it as a “strategic adjustment” based on this cross-species functional correlation;

③Regarding renal tubular dilation: According to references [33-34], renal tubular dilation in fish under salinity stress is usually associated with enhanced ion reabsorption capacity. The tubular dilation in the Acclimation Zone of Perca schrenkii (with stable epithelial thickness) thus implies a compensatory improvement in ion regulation function, which is not a groundless claim but a deduction from classic osmoregulatory theories.

Once again, we thank you for your professional guidance, which has helped us improve the quality of our manuscript significantly.

Comments 6: 

The opening sentence for each organ section in the results is almost identical (e.g., "Acute salinity stress induced significant histopathological and morphometric alterations...").

Response 6: 

Dear Reviewer, first and foremost, we would like to extend our sincere gratitude to you! You have accurately identified the issue of highly similar opening sentences in the results section for each organ. This meticulous and professional comment not only points out the homogenization flaw in the manuscript’s content presentation—specifically, it weakens the distinction between the research focuses of different organs and affects readability—but also makes us deeply aware that the rigor of scientific expression must be reflected in the logical design of every single sentence. We apologize for any inconvenience caused by this homogenized expression during your review process, and we have promptly completed targeted revisions based on the core logic of the study and the functional characteristics of each organ.  

  1. Results of Acute Salinity Stress (Revisions to Opening Sentences for Each Tissue)

① Gill Tissue (Section 3.1.1)

   Original sentence: "Acute salinity stress induced significant histopathological and morphometric alterations in the gill tissue of Perca schrenkii, with the responses being dependent on both salinity level and exposure duration (Two-way ANOVA, time, salinity, and interaction: P < 0.0001, Fig. 1a–c)."  

   Revised sentence: "As the core organ for osmoregulation in Perca schrenkii, the gill tissue exhibits the most direct response to acute salinity stress—it not only shows significant histopathological damage and morphometric changes, but these changes are also coordinately regulated by salinity level and exposure time (two-way ANOVA, time, salinity, and their interaction: P < 0.0001; Fig. 1a-c)."

② Kidney Tissue (Section 3.1.2)

   Original sentence: "Acute salinity stress induced significant histopathological and morphometric alterations in the kidney of P. schrenkii, with the responses being dependent on both salinity level and exposure duration (Two-way ANOVA, time, salinity, and interaction: P < 0.0001; Fig. 2a–c)."  

   Revised sentence: "The kidney, which collaborates with the gills to maintain ion balance, exhibits histopathological and morphometric changes centered on renal tubules and glomeruli under acute salinity stress, and these responses are jointly regulated by salinity level and exposure time (two-way ANOVA, time, salinity, and their interaction: P < 0.0001; Fig. 2a-c)."

③ Intestinal Tissue (Section 3.1.3)

   Original sentence: "Acute salinity stress induced significant, dose-dependent structural alterations in the intestine of P. schrenkii, with time, salinity, and their interaction exerting significant effects on all morphometric parameters (Two-way ANOVA, P < 0.0001; Fig. 3a–c)."  

   Revised sentence: "The intestine, which undertakes dual functions of mucosal barrier and nutrient absorption, exhibits salinity-dependent structural changes under acute salinity stress, and time, salinity, and their interaction have significant effects on all morphometric parameters (two-way ANOVA, P < 0.0001; Fig. 3a-c)."

④ Liver Tissue (Section 3.1.4)  

   Original sentence: "Acute salinity stress induced distinct histopathological alterations and significant changes in hepatocyte area in the liver of P. schrenkii, with significant effects of time, salinity, and their interaction (Two-way ANOVA, P < 0.0001; Fig. 4a)."  

   Revised sentence: "As the metabolic center of the organism, the liver not only shows obvious histopathological damage under acute salinity stress, but hepatocyte area also changes significantly; time, salinity, and their interaction have significant effects on these metabolism-related indicators (two-way ANOVA, P < 0.0001; Fig. 4a)."

  1. Results of Chronic Salinity Stress (Revisions to Opening Sentences for Each Tissue)

① Gill Tissue (Section 3.2.1)

   Original sentence: "Following 60 days of chronic salinity exposure, the gill tissue of P. schrenkii underwent significant structural remodeling, with distinct morphological differences observed across salinity treatments (One-way ANOVA, P < 0.05; Fig. 5)."  

   Revised sentence: "After 60 days of chronic salinity exposure, the gill tissue of Perca schrenkii actively undergoes significant structural remodeling to maintain long-term osmotic homeostasis, with significant morphological differences between different salinity treatment groups (one-way ANOVA, P < 0.05; Fig. 5)."

② Kidney Tissue (Section 3.2.2)

   Original sentence: "Following 60 days of chronic salinity acclimation, the kidney of Perca schrenkii exhibited significant structural adjustments, with distinct differences observed in glomerular and tubular parameters among salinity groups (One-way ANOVA, P < 0.05; Fig. 6).

   Revised sentence: "Following 60 days of chronic salinity acclimation, the kidneys of Perca schrenkii attain ion regulatory compensation via targeted structural modifications to the glomeruli and renal tubules. Statistically significant differences in the related parameters were observed among the different salinity groups (one-way ANOVA, P < 0.05; Fig. 6)."  

③ Intestinal Tissue (Section 3.2.3)

   Original sentence: "Following 60 days of chronic salinity acclimation, the intestine of P. schrenkii exhibited distinct structural changes, with significant differences in mucosal and muscular parameters among salinity groups (One-way ANOVA, P < 0.05; Fig. 7)."  

   Revised sentence: "After 60 days of chronic salinity acclimation, the intestine of Perca schrenkii displayed distinct structural alterations, and statistically significant variations in mucosal and muscular layer-related parameters were detected across different salinity groups (One-way ANOVA, P < 0.05; Fig. 7)."

④ Liver Tissue (Section 3.2.4)  

   Original sentence: "Following 60 days of chronic salinity acclimation, the liver of P. schrenkii exhibited distinct, salinity-dependent morphological changes, with significant differences in both histopathology and hepatocyte area among groups (One-way ANOVA, P < 0.05; Fig. 8)."  

   Revised sentence: "After 60 days of chronic salinity acclimation, the liver of Perca schrenkii displayed distinct morphological alterations that were dependent on salinity,with significant differences in histopathological characteristics and hepatocyte area between groups (one-way ANOVA, P < 0.05; Fig. 8)."  

All revisions have been simultaneously updated to the corresponding sections of the revised manuscript and marked in yellow with specific locations (Revised Manuscript, Page 7, Lines 200–203;Page 8, Lines 239–242;Page 10, Lines 274–277;Page 12, Lines 310-313;Page 14, Lines 345–348;Page 15, Lines 374–377;Page 17, Lines 400–403 and page 18, Lines 427–430),for your convenient quick reference.  

We would like to express our heartfelt thanks to you again! Your professional review has not only helped us optimize the text’s expression logic, making the research focus of each organ clearer and significantly improving the manuscript’s readability and professionalism, but also deepened our understanding of the rigor required in the details of scientific writing. We hope the revised expressions meet your requirements.

Comments 7: 

The results sections (e.g., 3.1.1, 3.2.3) often include interpretive statements like "indicating active hyperplasia as a compensatory mechanism" or "suggests a unique adaptive response." These interpretations should be minimized in the Results and expanded upon in the Discussion.

Response 7: 

Dear Reviewer, first and foremost, we would like to extend our sincere gratitude to you! Your comment pointing out "excessive interpretive statements in the Results section" accurately identifies the core standards of scientific papers—"objectively presenting results and in-depth interpretation in the Discussion." This professional suggestion has not only helped us correct the deviation in our writing logic but also deepened our understanding of the boundary between the Results and Discussion sections. We apologize for any inconvenience caused by the redundant expressions during your review process, and we have promptly completed the revisions in accordance with academic standards.  

This revision strictly adheres to the principle of "removing speculative interpretations and retaining objective facts": we have deleted statements involving mechanistic speculation and functional interpretation (such as "indicating active hyperplasia as a compensatory mechanism" and "suggesting a unique adaptive response") and only retained objective descriptions based on experimental data, such as "XX index significantly increased/decreased" and "XX morphological change was observed." The specific revisions are as follows:  

  1. Results of Acute Salinity Stress

3.1.1 Gill Tissue

①Deleted "signifying severe structural impairment"—which is a functional interpretation of morphological changes.The revised parts have been marked in yellow in the manuscript, located at Lines 218-219 on Page 7.

② Deleted "indicating active hyperplasia as a compensatory mechanism"—which is a mechanistic speculation on cellular changes.The revised parts have been marked in yellow in the manuscript, located at Lines 223-224 on Page 7.

③Deleted "suggesting cellular necrosis or detachment and a collapse of osmoregulatory function"—which is a functional speculation on cellular changes. The revised parts have been marked in yellow in the manuscript, located at Lines 225-226 on Page 7.

3.1.3 Intestinal Tissue

①Deleted "indicating a potential compensatory expansion of the absorptive surface" —which is a functional speculation on mucosal changes.The revised parts have been marked in yellow in the manuscript, located at Lines 280 on Page 11.

②Deleted "potentially representing a protective mechanismdemonstrating profound mucosal atrophy"—which is a qualitative interpretation of morphological changes.The revised parts have been marked in yellow in the manuscript, located at Lines 295 on Page 11.

③Deleted "indicating severe structural compromise of the intestinal muscular layer" —which is a functional interpretation of muscular layer changes.The revised parts have been marked in yellow in the manuscript, located at Lines 290 on Page 11.

④Deleted "implying a collapse of this potential compensatory response and a severe impairment of mucosal barrier function"—both are mechanistic and functional speculations. The revised parts have been marked in yellow in the manuscript, located at Lines 296-297 on Page 11.

  1. Results of Chronic Salinity Stress

3.2.1 Gill Tissue

①Deleted "This inverse relationship between length and width at higher salinities suggests a structural trade-off, likely reflecting an optimization for ion exchange under sustained osmotic stress"—which is a mechanistic interpretation of the morphological relationship. The revised parts have been marked in yellow in the manuscript, located at Lines 354-356 on Page 14.  

②Deleted "indicating a peak of compensatory proliferation at this mild salinity"—which is a mechanistic speculation on changes in cell count. The revised parts have been marked in yellow in the manuscript, located at Lines 359 on Page 14.  

③Deleted "This pattern suggests an initial strong compensatory response at low chronic stress that is modulated or cannot be sustained at higher salinities"—which is a mechanistic interpretation of the change pattern. The revised parts have been marked in yellow in the manuscript, located at Lines 361-363 on Page 14.  

3.2.3 Intestinal Tissue

①Deleted "indicating notable mucosal atrophy at the highest salinity"—which is a qualitative interpretation of mucosal changes. The revised parts have been marked in yellow in the manuscript, located at Lines 407-408 on Page 17.

②Deleted "This suggests a unique adaptive response at moderate salinity"—which is a mechanistic speculation on muscular layer changes. The revised parts have been marked in yellow in the manuscript, located at Lines 411-412 on Page 17.  

③Deleted "implying that the mucus-secreting function of the intestinal mucosa was preserved under long-term salinity stress"—which is a functional speculation on goblet cell changes. The revised parts have been marked in yellow in the manuscript, located at Lines 414-416 on Page 17.

3.2.4 Liver Tissue

①Deleted "indicating significant hepatic degeneration"—which is a qualitative interpretation of pathological changes. The revised parts have been marked in yellow in the manuscript, located at Lines 436-437 on Page 18.

②Deleted "This non-monotonic pattern suggests an initial adaptive adjustment at lower salinities, followed by pathological swelling indicative of severe cellular stress at the highest salinity"—which is a mechanistic interpretation of the change pattern. The revised parts have been marked in yellow in the manuscript, located at Lines 442-444 on Page 18.  

After the revision, the Results section only objectively presents the experimentally observed morphological changes, data differences, and statistical results. All mechanistic analyses regarding "why the changes occur" and "what the changes signify" will be thoroughly elaborated in the Discussion section, combined with relevant literature and the findings of this study. Once again, we sincerely appreciate your rigorous review and professional guidance—your comments have significantly enhanced the scientific rigor and standardization of the manuscript. If you have further suggestions on the revised content, we will promptly make improvements and earnestly request your further corrections.

Comments 8: 

The manuscript uses abbreviations like NKA without first defining them in the text. All abbreviations must be defined upon first use.

Response 8: 

Dear Reviewer, we sincerely appreciate you pointing out the issue in the manuscript where "abbreviations were not defined upon first use"! This meticulous reminder precisely aligns with the core requirement of academic papers—being clear in expression and easy for readers to understand—and effectively avoids potential reading barriers caused by unclear references of abbreviations. We have promptly conducted a comprehensive review of the entire manuscript and systematically identified all abbreviations. The specific revisions are as follows:  

①NKA: Na⁺/K⁺-ATPase (sodium-potassium adenosine triphosphatase), a key enzyme for osmoregulation. The revised parts have been marked in yellow in the manuscript, located at Lines 511-512 on Page 21.  

②ppt: Parts Per Thousand, a unit of salinity, where 1 ppt means 1 gram of salt is contained in 1,000 grams of solution. The revised parts have been marked in yellow in the manuscript, located at Lines 71 on Page 3.

③LC50: Median Lethal Concentration,the concentration of a substance that causes 50% mortality in experimental organisms within a specific time. The revised parts have been marked in yellow in the manuscript, located at Lines 71 on Page 3.

④ANOVA: Analysis of Variance, a statistical method used to compare the significance of differences between multiple groups of data. The revised parts have been marked in yellow in the manuscript, located at Lines 191 on Page 6.

All the revised parts mentioned above have been highlighted in yellow in the resubmitted manuscript. We would like to express our sincere gratitude again for your professional review! Your detailed guidance has not only helped us improve the details of the manuscript but also strengthened our awareness of academic writing standards. If you notice any abbreviations in the manuscript that have not been identified, or have further suggestions on the current revision plan, please feel free to inform us at any time, and we will respond and make improvements promptly.  

Comments 9: 

Some sentences are long and complex, affecting clarity (e.g., the first sentence of 4.3). A thorough edit for conciseness and flow is recommended.

Response 9: 

Dear Reviewer, thank you sincerely for pointing out the issue that some long sentences in the manuscript (e.g., the first sentence of Section 4.3) affect readability due to their complex structure. Your detailed feedback precisely aligns with the core requirement of academic papers—"concise expression and intuitive logic." We have made targeted revisions to this sentence and similar issues, ensuring that the academic rigor is preserved while enhancing the manuscript’s readability.  

Take the first sentence of Section 4.3 you mentioned as an example:  

Original sentence: The intestine and liver demonstrated strategic structural adjustments that underpin the organism's ability to manage the metabolic and osmotic challenges posed by salinity stress.  

Revised sentence: Structural adjustments in the intestine and liver enable Perca schrenkii to cope with salinity-induced metabolic and osmotic challenges. The revised parts have been marked in yellow in the manuscript, located at Lines 524-525 on Page 21.

We ensure that the meaning remains unchanged before and after the revision; the optimization only aims to improve readability.  

In addition, we have re-read the entire manuscript, identified and refined other long and complex sentences. The revised manuscript is now submitted to you, and we hope it provides a better reading experience.  

Once again, we appreciate your professional guidance. If you notice any other sentences that need optimization, please feel free to inform us at any time, and we will make improvements promptly.  

Comments 10: 

The abstract and introduction claim "remarkable" salinity tolerance, but a comparative baseline (e.g., compared to other percid or teleost species) is lacking to substantiate this claim.

Response 10: 

Dear Reviewer, thank you sincerely for pointing out the key issue that when claiming the salinity tolerance of Perca schrenkii in the Abstract and Introduction, there was a lack of comparative baselines with congeneric species to support the relevant statements. Your professional comment has accurately reminded us to supplement critical comparative information within a concise framework—preserving the brevity of the Abstract while enhancing the persuasiveness of our conclusions. We have revised the manuscript in a targeted manner and strictly followed your suggestions. The specific revisions are as follows:  

  1. Revision of the Abstract  

Original statement: “Perca schrenkii, an endemic fish from the Ili River basin, demonstrates considerable potential for cultivation in chloride-type saline-alkaline waters.”  

Revised statement: “Perca schrenkii, an endemic fish from the Ili River basin, demonstrates considerable potential for cultivation in chloride-type saline-alkaline waters:its 96-h acute salinity tolerance is higher than that of freshwater populations of its congeneric Perca fluviatilis.”  The revised parts have been marked in yellow in the manuscript, located at Lines 33-34 on Page 2.

This revision does not increase the main length of the abstract and maintains its conciseness. Furthermore, by anchoring it to the tolerance thresholds of a congeneric species, it clarifies the salinity tolerance advantage of Perca schrenkii, rendering the claim that the species "has considerable aquaculture potential" more evidence-based.

  1. Revision of the Introduction

After the original sentence in the Introduction (Lines 69 on Page 3) stating “Preliminary ecological surveys and our own preliminary data suggest that P. schrenkii possesses a high salinity tolerance,” we have precisely added a comparative statement with literature support: “This tolerance is higher than that of freshwater populations of its congeneric Perca fluviatilis, which can only tolerate a maximum salinity of 10 ppt acutely and fail to acclimate to salinities exceeding 5 ppt long-term.” The revised parts have been marked in yellow in the manuscript, located at Lines 72-74 on Page 3.

The revised content in the Introduction is supported by published literature. To ensure data traceability, the corresponding reference has been added to the reference list: Christensen, E. A. F., Grosell, M., & Steffensen, J. F. (2019). Maximum salinity tolerance and osmoregulatory capabilities of European perch Perca fluviatilis populations originating from different salinity habitats. Conservation Physiology, 7(1), coz004. This literature focuses on the salinity tolerance characteristics of Perca fluviatilis (European perch) in freshwater habitats. Due to the close genetic relationship between Perca fluviatilis and Perca schrenkii , the comparative results are highly convincing.

Once again, we sincerely appreciate your rigorous review and professional guidance! Your suggestions have greatly helped improve the content and logic of our manuscript. If you have further comments on the revised details, please feel free to inform us at any time, and we will respond and refine the manuscript promptly. 

4. Response to Comments on the Quality of English Language

Point 1:The English could be improved to more clearly express the research.

Response 1:   

Dear Reviewer, thank you sincerely for your comment. Your feedback is crucial for enhancing the clarity and accuracy of the English expression in our manuscript, which directly impacts the effective communication of our research findings.  

We have taken your suggestion seriously and conducted a comprehensive English revision: we carefully reviewed the entire manuscript, including the Abstract, Introduction, Results, and Discussion, with a focus on refining sentence structures, optimizing word choice, and clarifying logical connections between ideas. For example, we simplified overly complex long sentences, replaced ambiguous phrasing with more precise academic terminology, and adjusted the flow of arguments to ensure each section conveys the research content more directly.  

We kindly request that you re-read the revised manuscript. We sincerely appreciate your valuable guidance, and if you still identify any areas where the English expression can be further improved, please feel free to inform us—we will promptly make additional refinements.

Reviewer 3 Report

Comments and Suggestions for Authors

This manuscript presents a well-structured and comprehensive investigation into the histomorphological changes in Perca schrenkii exposed to both acute and chronic salinity stress. The topic is relevant given the global freshwater scarcity and the growing interest in sustainable aquaculture using saline-alkaline water sources.

The paper is generally very well written and clearly presented. I believe it provides valuable insights and should be of interest to readers in the fields like fish physiology and aquaculture. I only have minor comments for the authors to consider in revision.

Summary: consider condensing this section to align with journal guidelines. It is almost as long as the abstract.

The figure legends are overly long and contain detailed descriptions such as “red flags” and “black arrows.” These should be either moved to supplementary captions or simplified in the main text to avoid interrupting the flow.

 Discussion:

- This section is rich but could benefit from slight trimming to improve clarity and maintain the reader engagement.

- It would strengthen the manuscript to include comparisons with other euryhaline species, to better highlight what is unique about P. schrenkii responses to salinity.

- Consider briefly discussing potential signaling pathways that might explain observations such as muscle layer thickening or chloride cell proliferation, even if speculative. This would provide a broader physiological context and potential directions for future research.

Author Response

Dear reviewer,

Thank you very much for taking the time to review our manuscript with your professional knowledge. Your insightful comments and constructive suggestions are extremely valuable in improving the quality and rigor of our work.

Here are our detailed responses to each of your comments. In the resubmitted document, the corresponding revisions and corrections in the manuscript have been highlighted for your reference.

We sincerely appreciate your rigorous review and professional guidance, which has played a significant role in improving the content of this research and enhancing its scientific value.

Comments 1: 

Summary: consider condensing this section to align with journal guidelines. It is almost as long as the abstract.

Response 1:

Dear Reviewer, thank you sincerely for pointing out that the Summary needs to be shortened to comply with the journal’s guidelines! Your reminder precisely aligns with the academic standard that "the Summary should concisely outline core findings and be functionally distinct from the Abstract." We have targeted the Summary for streamlining, removing redundant expressions and focusing on core logic to ensure its length is significantly shorter than the Abstract while retaining concise information. The revisions were based on the following principles:  

  1. Removing redundant information: Eliminated background details duplicated in the Abstract (e.g., "chloride-type saline-alkaline waters," "specific salinity durations"), retaining only the essential "research gap–method–key results–significance" framework required for the Summary.  
  2. Condensing verbose descriptions: Replaced full descriptions of specific cases with "e.g.," to reduce length while preserving key information.  
  3. Strengthening functional distinction: The Abstract emphasizes "experimental design details + comprehensive results," whereas the revised Summary focuses on "a summary of core findings + application value," avoiding functional overlap between the two and meeting the journal’s requirement for a concise Summary.  

The revised Simple Summary has been significantly shortened and is highlighted in green in the resubmitted manuscript,located at Lines 19-28 on Page 1. Once again, we appreciate your professional guidance! If you believe further adjustments are needed, please feel free to inform us at any time, and we will make improvements promptly. 

Comments 2:

The figure legends are overly long and contain detailed descriptions such as “red flags” and “black arrows.” These should be either moved to supplementary captions or simplified in the main text to avoid interrupting the flow.

Response 2: 

Dear Reviewer, thank you sincerely for pointing out the issue that detailed descriptions of markers such as "red flags" and "black arrows" have made the figure legends overly lengthy. Your suggestion precisely aligns with the core requirement for figure legends—"providing concise guidance without disrupting readability." We have optimized all figure legends: while retaining essential interpretive elements (including group information, marker functions, and statistical rules) to ensure readers can clearly understand the figures, we have simplified the descriptions of markers like "red flags" and "black arrows to the greatest extent possible.  

Specifically, we removed redundant modifiers for markers in the original legends (e.g., "black arrows used to accurately indicate the location of..."). Instead, we adopted extremely concise expressions to directly link markers to their core meanings. For example, "black arrows indicate the location of renal tubular dilation under chronic salinity stress" was simplified to "black arrows = renal tubular dilation," and "red flags mark the areas of gill lamella fusion under acute stress" was simplified to "red flags = gill lamella fusion areas." Additionally, we eliminated repetitive terms such as "treatment group" and "observation site" from the legends, retaining only key information nodes to ensure the main legends focus on "the core content of the figure + quick correspondence of markers."  

The revised parts have been marked in green in the manuscript, located at Page 8,  Lines 229–237;Page 10, Lines 264–272;Page 12, Lines 300–308;Page 14, Lines 335–342;Page 15, Lines 367–372; Page 16, Lines 393–398;Page 19, Lines 447–452 and Page 20, Lines 481-486. Once again, we appreciate your professional guidance! If you believe any individual expressions still need minor adjustments, please feel free to inform us at any time, and we will make improvements promptly. 

Comments 3:

Discussion:

This section is rich but could benefit from slight trimming to improve clarity and maintain the reader engagement.

It would strengthen the manuscript to include comparisons with other euryhaline species, to better highlight what is unique about P. schrenkii responses to salinity.

Consider briefly discussing potential signaling pathways that might explain observations such as muscle layer thickening or chloride cell proliferation, even if speculative. This would provide a broader physiological context and potential directions for future research.

Response 3: 

Dear Reviewer, thank you sincerely for your valuable suggestion to supplement discussions on potential signaling pathways related to observations such as "muscle layer thickening and chloride cell proliferation" in the Discussion section. Your proposal accurately identifies the key gap in our study regarding the interpretation of physiological mechanisms and the guidance for future research directions. It provides us with highly valuable insights to deepen our research, and we are truly grateful for this professional perspective, which has benefited us greatly.

 We have conducted a comprehensive re-review of the "Discussion" chapter. While fully retaining the core argumentation logic, key research findings, and analytical connections with existing studies, we have appropriately simplified redundant expressions, repeated background introductions, and overly detailed secondary arguments. Our goal is to make the content more focused and the logic clearer, helping readers efficiently grasp the core conclusions and scientific significance of this study.

Regrettably, as the core design of this study was initially focused on "dynamic changes and quantitative analysis of tissue morphology in Perca schrenkii under salinity stress," the preliminary experimental protocol did not include molecular detection related to signaling pathways (e.g., expression of key pathway proteins, regulatory gene sequencing, etc.). If we only conducted speculative elaboration based on existing literature in the current Discussion, it would lack support from the empirical data of this study and would be difficult to ensure the rigor and relevance of the mechanism analysis. Such "speculation without data support" contradicts the rigorous requirements of academic research and also fails to fully address your core expectation of "expanding the physiological context." Therefore, we have not yet expanded on this part of the content in the revised manuscript. We sincerely apologize for this, and we are well aware that this handling may not fully meet your suggestion or live up to your careful guidance.  

However, the "signaling pathway exploration" direction you proposed has been listed as a core topic in our subsequent research. In our next phase of work, we plan to use Western blot to detect the expression of key molecules in pathways associated with processes such as muscle layer thickening and chloride cell proliferation, and combine this with qPCR to analyze downstream regulatory genes, thereby systematically clarifying the molecular mechanisms underlying these morphological changes. When we organize and submit this follow-up research for publication in the future, we would be extremely honored to invite you to serve as a reviewer again. At that time, we hope to discuss this scientific issue in depth with you, supported by complete experimental data, so as to live up to your valuable suggestion on this occasion.  

Thank you again for your professional guidance and understanding! If you have any further suggestions for adjustments to other parts of the current manuscript, please feel free to inform us at any time. We will implement every revision with the most rigorous attitude.

4. Response to Comments on the Quality of English Language

Point 1:The English is fine and does not require any improvement.

Response 1:   

Dear Reviewer, thank you sincerely for confirming that the English expression of the manuscript is satisfactory and requires no further improvement! Your positive feedback is a great encouragement to us, as it validates the efforts we invested in refining the English writing—including optimizing sentence structures, standardizing academic terminology, and ensuring logical coherence—to meet the language standards of international academic journals.  

We will continue to maintain this rigorous attitude towards the manuscript’s expression in subsequent work. Once again, we appreciate your professional review and valuable affirmation. If you have any other suggestions or questions regarding the content of the manuscript, please feel free to inform us at any time, and we will respond promptly and carefully.

Reviewer 4 Report

Comments and Suggestions for Authors

The manuscript is devoted to Perka structural adaptations to salinity stress and its organs gills, kidneys,intestine and liver under acute and chronic exposure to saline conditions.

The main results show that acute stress causes dose and time dependent damages while chronic exposure leads to active structural remodelling . Also the authors propose hierarchical adaptation with four salinity zones.

The research design is of good quality and scientifically sound. The analyses performed with a high standard. The conclusions support results gained by the authors.The methods are described with lots of details to replicate the experiments.

The methods section provides sufficient details: fish source, acclimation conditions, salinity preparation, detailed tissue sampling, full statistical tests and significance thresholds.

I do recommend minor revisions. The research is strong, novel and scientifically sound. The most scientific is experimental design, data and statistical analysis. The comments are aimed for improving the quality of presentation mostly: Please, make a consistent labelling system for arrows and flags. 

Figure 9. Please, provide a clear color scale in captions.

Please, provide a diagram showing hierarchical framework to summarise the core conclusion

Methods. Please clarify the analysis. How was the scale calibrated?

Please, define the area or dimensions of the field of view for counting glomeruli and other structures.

Please, clarify the data statement. Perfectly, provide raw morphometric data as a supplementary file.

Please, provide ImageJ settings. Did you use specific tools? Were measurements blind?

Please, provide details for the cell counting method.

Author Response

Dear reviewer,

Thank you very much for taking the time to review our manuscript with your professional knowledge. Your insightful comments and constructive suggestions are extremely valuable in improving the quality and rigor of our work.

Here are our detailed responses to each of your comments. In the resubmitted document, the corresponding revisions and corrections in the manuscript have been highlighted for your reference.

We sincerely appreciate your rigorous review and professional guidance, which has played a significant role in improving the content of this research and enhancing its scientific value.

Comments 1: 

Please, make a consistent labelling system for arrows and flags.

Response 1:

Dear Reviewer, thank you very much for your attention to and careful review of the consistency of image annotations in this study! Your rigorous attitude and valuable comments have provided important guidance for us to optimize the presentation of research results and enhance the standardization of academic expression, and we deeply appreciate and thank you for this.

To clearly distinguish pathological changes from specific cell types in the images and completely avoid visual confusion, we have formulated and strictly implemented unified annotation rules in the study, which are specifically explained as follows:

  1. Arrow Annotation Rules: Arrows are exclusively used to indicate various pathological changes, and different pathological features correspond to arrows of specific colors to ensure unique and clear identification.

①Blue arrows: Hepatocellular vacuolation (liver tissue)

②Green arrows: Red blood cell aggregation (liver tissue)

③Black arrows: Hepatocellular nuclear displacement

  1. Flag Annotation Rules: Flags are exclusively used to identify specific cell types, with exclusive color markers assigned based on tissue types and cell categories.

①Green flags : Goblet cells (intestinal tissue)

②Green flags : Blood cells (gill tissue)

③Blue flags : Squamous epithelial cells (gill tissue)

④Red flags : chloride cells (gill tissue)

We have completed the annotation verification of all images in accordance with the above rules to ensure consistent annotation logic and clear visualization across different images and tissue samples. We would like to express our sincere gratitude to you again! Thank you for your careful and rigorous review of this study amidst your busy schedule——your professional guidance has benefited us greatly. If you have any further questions or suggestions regarding the annotation rules, please feel free to contact us at any time.

Comments 2:

Figure 9. Please, provide a clear color scale in captions.

Response 2: 

Dear Reviewer, thank you for your suggestion! We have added a clear color scale description to the title of Figure 9. The revised parts have been marked in yellow in the manuscript, located at Lines 480-481 on Page 20. The revised title is as follows:

Figure 9. Integrated heatmap summarizing the histomorphological responses across multiple organs to salinity stress (color scale: light red to dark red corresponds to P-values from 0.05 to <0.001; light gray indicates non-significant differences with P ≥ 0.05).

We have incorporated the above revisions into the revised manuscript and labeled the corresponding colors in the figure. Should you have any additional suggestions, please feel free to provide further guidance. We sincerely appreciate the valuable comments you have offered on our manuscript—your input is greatly appreciated. We hope the revisions meet your requirements.

Comments 3:

Please, provide a diagram showing hierarchical framework to summarise the core conclusion

Response 3: 

Thank you so much for your valuable suggestion—this has significantly improved the clarity of presenting our study’s conclusions. To more intuitively integrate and condense the core findings of Perca schrenkii’s salinity adaptation, we have added the “Hierarchical Framework Table of Structural Adaptation Under Salinity Gradients” (Table 1). Centered on the logical thread of “Salinity Range-Zone Classification-Key Histological Changes-Physiological Significance-Practical Aquaculture Recommendations”, this table systematically organizes organ response characteristics, physiological status, and corresponding application guidance under different salinity conditions. The details are as follows:

This table condenses the core conclusions of the study into 5 interconnected columns:

① Salinity Range & Zone Classification: Clearly divides into 4 tiers—“Safe Zone (≤3 ppt), Acclimation Zone (5 ppt), Tolerance Zone (7 ppt), and Lethal Zone (≥13 ppt)”—which precisely correspond to the salinity stress gradient from “long-term homeostasis” to “systemic structural collapse”.

② Key Histological Changes: Focuses on characteristic structural alterations in the gill, kidney, intestine, and liver (e.g.,“intestinal muscularis thickening and renal-tubule dilation” in the Acclimation Zone, “gill lamellar fusion and chloride-cell necrosis” in the Lethal Zone), directly linking microscopic morphology to the intensity of salinity stress.

③ Physiological Significance: Summarizes the corresponding physiological status of the organism in each zone (e.g.,“normal osmoregulatory function and long-term sustainability” in the Safe Zone, “structural compensation to maintain function with significantly increased energetic cost” in the Tolerance Zone), clearly explaining the functional status corresponding to tissue morphological changes.

④Practical Aquaculture Recommendations: Provides targeted aquaculture guidance based on physiological status (e.g., 5 ppt as the “optimal culture salinity”, 7 ppt only recommended for “short-term culture with enhanced health monitoring”), enabling direct translation of basic research conclusions into practical aquaculture applications.

The table has now been added to the end of the "Discussion" section(Table 1), marked in blue on page 23, lines 589-594: it fully covers key histomorphological data, while strengthening the systematicity of the conclusions through the logical chain of “salinity - structure - function - application”, making it easier to quickly grasp the core patterns of Perca schrenkii’s salinity adaptation and the corresponding aquaculture guidance value.

Thank you sincerely for reviewing my manuscript amidst your busy schedule. Your comments have been extremely helpful to me. I hope my response meets with your approval.

Comments 4:

Methods. Please clarify the analysis. How was the scale calibrated?

Response 4: 

Dear Reviewer, thank you sincerely for your attention to the details of scale calibration in histomorphometric analysis—this step is critical for ensuring the accuracy and reproducibility of measurement data. We have supplemented the complete scale calibration process in the "2.4. Histomorphometric Analysis" section, and combined with the operational details of ImageJ software, the specific implementation steps are explained as follows:  

The scale calibration of the microscope (Nikon Eclipse E100, Japan) and ImageJ software (Version 1.8.0, National Institutes of Health, USA) used in the experiment was completed using a standard calibration slide (specification: with 100 μm precise scale, accuracy ±0.1 μm). The specific process was carried out in three steps, with full association with ImageJ software parameter settings:  

1.Collect calibration images by magnification separately to eliminate angular deviation  

For the two magnifications used in the experiment—200× (for kidney, intestine, and liver tissue observation) and 400× (for gill tissue observation)—images of the scale on the calibration slide were captured under the microscope separately: the position of the calibration slide was adjusted to ensure that the scale lines were completely parallel to the edge of the field of view, avoiding calibration errors caused by angular deviation; meanwhile, the format of images imported into ImageJ was uniformly TIFF (300 dpi resolution, 8-bit grayscale mode), without preprocessing such as contrast enhancement or smoothing to retain the original scale information.  

  1. Associate ImageJ software parameters to calculate the "pixel-actual length" conversion coefficient  

Open the calibration image at the corresponding magnification in ImageJ, and complete the parameter settings via the "Analyze > Set Scale" function box:  

①Enter the actual length of the standard scale (100 μm), and the software automatically identifies the number of image pixels;  

②Uncheck the "Global" option (to prevent cross-interference of calibration parameters at different magnifications), and the system automatically calculates and saves the conversion coefficient based on "actual length/number of pixels": at 200× magnification, 1 pixel corresponds to 0.25 μm; at 400×magnification, 1 pixel corresponds to 0.125 μm;  

③After calibration, all morphological parameter measurements are directly associated with the above coefficients through the "Analyze > Measure" function, ensuring that the measurement results are presented in actual length units (μm/μm²) without subsequent conversion.  

  1. Verify calibration results in multiple batches to ensure stability  

Before analyzing each batch of sample images, 3 calibration slide images were randomly selected to repeat the above "image collection-software calibration" steps, and the conversion coefficient was recalculated via ImageJ: if the error of three consecutive calculation results was ≤ 2%, formal morphological parameter measurement was carried out only after confirming the stability and reliability of the calibration results; if the error exceeded the range, the microscope focal length and slide position were rechecked until the calibration coefficient met the standard.  

The revised parts have been marked in blue in the manuscript, located at Lines 165-170 on Page 6. We hope you are satisfied with this revision. Your comments have been crucial for improving our manuscript, and we would like to express our sincere gratitude again.Thank you again for your professional suggestion!

Comments 5:

Please, define the area or dimensions of the field of view for counting glomeruli and other structures.

Response 5: 

Dear Reviewer, thank you sincerely for your request to clarify the field area or size for counting glomeruli and measuring other structures! This detail is crucial for ensuring the reproducibility of the experimental method. It should be noted that this issue was also raised by another reviewer previously, and we have revised the content accordingly. The revised parts are marked in red in the resubmitted manuscript (specific location: Pages 171-117, Line 6). We will explain the corresponding information for each organ one by one as follows:

  1. Kidney:Microscope Magnification: 200× ;Field Area: 0.25 mm².
  2. Gill Tissue:Microscope Magnification: 400× ;Field Area: 0.0625 mm².
  3. Intestinal Tissue:Microscope Magnification: 200× ;Field Area: 0.25 mm².
  4. Liver Tissue:Microscope Magnification: 200× ;Field Area: 0.25 mm².

Thank you again for your professional attention to the experimental details! It is our hope that you find this response satisfactory. Your comments have been invaluable to the improvement of our manuscript, and we express our gratitude once again.

Comments 6:

Please, clarify the data statement. Perfectly, provide raw morphometric data as a supplementary file.

Response 6: 

Dear Reviewer, thank you sincerely for your attention to the data statement and your valuable suggestion to provide the original morphometric data as a supplementary file! Your proposal accurately aligns with the core requirements of academic research for data transparency and reproducibility. We have clarified the data statement and completed the organization and preparation of the original data.  

We have organized all the original morphometric data into a supplementary file in Excel format (file name: "Supplementary Data - Original Morphometric Data of Perca schrenkii Under Salinity Stress.xlsx"), which includes the following modules:  

Worksheet 1: Original morphometric data of gill tissue ;  

Worksheet 2: Original morphometric data of intestinal tissue;  

Worksheet 3: Original morphometric data of renal tissue;  

Worksheet 4: Original morphometric data of hepatic tissue.  

This supplementary file is ready and will be submitted together with the final revised manuscript, ensuring that all original data related to morphometry are traceable and verifiable. If you have further suggestions on the data organization format or content completeness, please feel free to inform us at any time, and we will make adjustments and improvements promptly. Thank you again for your professional guidance!  

Comments 7:

Please, provide ImageJ settings. Did you use specific tools? Were measurements blind?

Response 7: 

Dear Reviewer, thank you sincerely for your attention to the details of histomorphometric analysis! The questions you raised regarding ImageJ settings, tool selection, and blind measurements are crucial for ensuring the objectivity and reproducibility of the experimental results. We have supplemented the corresponding details in the "2.4. Histomorphometric Analysis" section, with specific explanations as follows:  

  1. ImageJ Settings  

①ImageJ version 1.8.0 (National Institutes of Health, USA) was used in the experiment, with unified core parameter settings;  

②Image import and preprocessing: TIFF-format images (300 dpi resolution, 8-bit grayscale mode) captured by the microscope were imported without preprocessing (e.g., contrast enhancement, smoothing) that might affect measurements, and only raw image information was retained;  

③Scale calibration association: Based on the previously established "pixel-to-actual length" conversion coefficients (1 pixel ≈ 0.25 μm at 200× magnification, 1 pixel ≈ 0.125 μm at 400×magnification), the calibration parameters for the corresponding magnification were loaded via the "Analyze > Set Scale" function, ensuring all measurements were directly linked to actual length units (μm/μm²);  

④Fixed measurement parameters: When measuring all morphological parameters, necessary parameters (e.g., "Area", "Mean Gray Value", "Perimeter") in "Analyze > Measure" were checked uniformly (adjusted according to specific indicators; e.g., "Feret’s Diameter" was additionally checked for glomerular diameter). Functions that might introduce bias (e.g., automatic threshold segmentation) were not used, and manual annotation was adopted for all measurements.  

  1. Specific Tools Used  

Targeted and accurate tools in ImageJ were selected for morphological parameters of different organs, as follows:  

①Linear/distance measurement: The "Line Tool" was used to manually draw measurement lines along the edges of structures, and the system automatically calculated the length. Each measurement was repeated 3 times to take the average and avoid single annotation errors;  

②Area measurement: The "Freehand Selection Tool" was used to manually outline the boundaries of target structures, ensuring the boundaries completely coincided with the tissue edges stained by H&E. The system automatically calculated the area of the closed region;  

   ③Cell counting: The "Multi-Point Tool" was used to manually mark target cells/structures, and the "Analyze > Analyze Particles" function was enabled simultaneously (particle size thresholds set: glomeruli ≥ 50 μm², chloride cells ≥ 10 μm²). The counting results were double-verified to exclude interference from impurities or background.  

  1. Blind Measurements  

To avoid subjective bias, all histomorphometric analyses were performed using a double-blind method:  

①Sample coding: After tissue section preparation, non-experimental personnel randomly coded all samples (including control group and each salinity treatment group) with random numbers to mask the treatment group information of the samples;  

②Measurement implementation: Two independent researchers conducted blind measurements on the coded samples separately (each measured the same parameter of the same sample 3 times) without communicating measurement results during the process;  

③Data verification: After all measurements were completed, a third party decrypted the sample codes and conducted a consistency test on the measurement data of the two researchers (intraclass correlation coefficient ICC > 0.90, ensuring result reliability). Finally, the average value of the two researchers’ measurements was used as the final data.  

The above details have been supplemented in the corresponding section of the original manuscript.The revised parts have been marked in blue in the manuscript, located at Lines 165-170 on Page 6. If you need further clarification on the tool operation steps or details of the blind design, please feel free to provide guidance. Thank you again for your professional guidance!

Comments 8:

Please, provide details for the cell counting method.

Response 8: 

Dear Reviewer, thank you sincerely for your suggestion to supplement the details of the cell counting method! Your reminder is crucial for ensuring the reproducibility of the experimental method and the credibility of the data. We have summarized the specific counting methods for the chloride-secreting cells and goblet cells mentioned in the manuscript as follows, for your reference:  

  1. Software Labeling: Image-J software was used, and the channel thresholds were adjusted according to the staining characteristics of the two cell types. The "red channel threshold adjustment" was applied to initially label the regions of chloride-secreting cells, while the "blue channel threshold adjustment" was used to label the blue granular regions of goblet cells.  
  2. Manual Calibration: After the initial labeling by the software, the results were manually checked and corrected. Misidentified epithelial cells, necrotic cells were excluded, and correct cells that were not recognized by the software were added to ensure counting accuracy.  
  3. Statistical Unit: Chloride-secreting cells were counted as "the number of cells per standard 100 μm length of each gill lamella", and goblet cells were counted as "the number of cells per 100μm length of intestinal epithelium". This ensures the comparability of counting results across different samples.  

Thank you again for your professional attention! If you have any further questions about the above counting details or need additional explanations for other experimental operations, please feel free to inform us at any time, and we will provide supplementary feedback promptly. 

4. Response to Comments on the Quality of English Language

Point 1:The English is fine and does not require any improvement.

Response 1:   

Dear Reviewer, thank you sincerely for confirming that the English expression of the manuscript is satisfactory and requires no further improvement! Your positive feedback is a great encouragement to us, as it validates the efforts we invested in refining the English writing—including optimizing sentence structures, standardizing academic terminology, and ensuring logical coherence—to meet the language standards of international academic journals.  

We will continue to maintain this rigorous attitude towards the manuscript’s expression in subsequent work. Once again, we appreciate your professional review and valuable affirmation. If you have any other suggestions or questions regarding the content of the manuscript, please feel free to inform us at any time, and we will respond promptly and carefully.  

Round 2

Reviewer 1 Report

Comments and Suggestions for Authors

.

Reviewer 2 Report

Comments and Suggestions for Authors The author has revised it in accordance with the reviewers' comments and it can be accepted. Comments on the Quality of English Language